# MODEL-FREE REINFORCEMENT LEARNING WITH NOISY ACTIONS FOR AUTOMATED EXPERIMENTAL CONTROL IN OPTICS

## ABSTRACT

Setting up and controlling optical systems is often a challenging and tedious task. The high number of degrees of freedom to control mirrors, lenses or phases makes automatic control challenging, especially when the complexity of the system cannot be adequately modeled due to noise or non-linearities. Here, we show that reinforcement learning (RL) can overcome these challenges when coupling laser light into an optical fiber, using a model-free RL approach that trains directly on the experiment without pre-training. By utilizing the sample-efficient algorithms Soft Actor-Critic (SAC) or Truncated Quantile Critics (TQC), our agent learns to couple with 90% efficiency, comparable to the human expert. We demonstrate that direct training on an experiment can replace extensive system modeling. Our result exemplifies RL's potential to tackle problems in optics, paving the way for more complex applications where full noise modeling is not feasible.

## 1 INTRODUCTION

In experimental physics, we work with complex and sensitive setups. Working in an optics lab means adjusting numerous mirrors, lenses, and other optical elements while optimizing complex parameters. Two of the main challenges are precision and the number of degrees of freedom. Often, tasks have to be repeated frequently. One example of such a task is coupling laser beams into optical fibers, used in many physics labs [1–4]. It can be a laborious and time-consuming task, especially in experiments with many fibers. Automating tasks like this can, therefore, free up domain expertise for more challenging tasks. Most of these repeated tasks have a very clear goal and can either be described as alignment or control problems. Alignment means the correct steering of a laser beam through an optical setup. Control refers to maintaining a dynamic experiment at a desired position using feedback loops. While fiber coupling is primarily an alignment problem, correcting for drift can be considered control.

Automation of alignment and control tasks is a classic use case of reinforcement learning (RL) [5–7]. RL has seen considerable success in recent years, both in general [8–14] and specifically in robotics [15–19]. However, due to many RL algorithms relying on a huge amount of data, at least in environments with continuous action spaces, most of these were performed in simulated or toy environments [20; 21]. Comparatively few experiments were done in real-world environments [22–24]. With the recent advance of more sample-efficient algorithms for continuous action spaces, like Deep Deterministic Policy Gradient (DDPG) [25], Twin Delayed Policy Gradient (TD3) [26], Soft Actor-Critic (SAC) [27], and Truncated Quantile Critics (TQC) [28], directly training in an experiment has become more feasible. However, we still face several challenges, such as partial observability, time-consuming training, and noise, when applying RL to real-world setups [22; 23].

In this work, we demonstrate how an RL agent successfully learns to couple light into an optical fiber, reaching efficiencies comparable to those of a human expert. We set up an experiment for fiber coupling on an optical table, motorizing the mirrors that guide the laser beam into the fiber. Our goal is to reach a specific coupling efficiency, which is the fraction of light entering the fiber. We have not included the absolute motor positions in the

observation in order to train our agent to improve the coupling efficiency for any type of misalignment, thus making the problem partially observable.

The primary issue we encountered was the lack of precision in the motors. For instance, returning to a position was only possible with a considerable and unpredictable offset, which leads to noise in the actions, a special type of stochasticity of the environment. In contrast to adding artificial noise to the actions for exploration [29–33], in our case, the noisy actions are inherent to the system. To solve this problem without a full analysis and modeling of the behavior, we let our agent train directly on the experiment with the standard StableBaselines3 [34] implementations of SAC and TQC. To reset a training episode, we could not reliably move to an absolute position but implemented a reset procedure that mainly relied on relative movement steps.

Despite the noisy actions and partial observability of the environment, the agent learns to reliably couple to an efficiency of $\geq 90\% \pm 2\%$ starting from a low power over the course of nearly four days. If we only need a smaller efficiency, e.g., $87\% \pm 2\%$, the training only takes twenty hours and can be performed in two nights, not taking away experimenting time. For comparison, the maximum coupling efficiency observed by the experimenter was 92%, and the one reached by the agent was 93%. We find that tuning the training parameters thoroughly is crucial to reducing training time, which is of high priority for real-world RL applications.

Our successful training is a first step towards further applications. First, our experiment shows that laser beam alignment using RL is generally possible. A transfer to other scenarios, such as interference optimization of two beams at a beam splitter or alignment of a laser field to an optical resonator, is straightforward and requires only a change of sensor [35]. Secondly, with training directly on the experiment, we show an example of applying RL to control tasks without having to model the experiment in detail beforehand. This is particularly important for more complicated experiments, such as those in quantum and atomic optics, where it may be disproportionate or impossible to simulate the exact dynamics and noise.

## 2 Related work

The application of RL to optical systems ranges through a wide range of topics including optical networks [36–44], adaptive optics [45–52], optical nanostructure, thin films and optical layers [53–56]. More related to our problem is work that studies how RL can be used to align and control tabletop optical experiments with lasers. In this category, some works are realized merely on simulation. Examples include studying mode-locked lasers [57], combining laser beams [58], and stacking laser pulses [59; 60]. Other works include investigating how RL performs in an actual experiment. Most of these studies, however, do not train on the experiment but on simulation. Examples include aligning an optical interferometer [61; 62], operating optical tweezers [63], combining laser beams [64] and operating pulsed lasers [65]. It is rare that the agent is trained directly on the experiment [24]. One example is a study combining pulsed laser beams [66]. Here, one actuator performs the actions, and the output is a scalar, the power. The training time is about 4 hours; simulations show that this would quickly go up to 1-2 days if more than two beams should be combined. Another example is the generation of a white light continuum [67]. Thereby, both the states and the actions are given by absolute positions of three actuators and moving to those positions, respectively. This gives their environment a relatively high observability for a real-world task. The authors claim to obtain successful training within 20 minutes. We deal with a higher (4) dimensional action space than both of these works. RL was also used to optimize the output power of an X-ray source [68]. For this, a single actuator was discretely controlled based on a scalar signal. As a side project, the paper looks at a simplified approach to fiber coupling using only two degrees of freedom and working with a discrete action space of size 4 employing DQNs [69]. The work does not present the achieved coupling efficiencies. In contrast, we work with continuous action spaces and control all four degrees of freedom necessary for general beam alignment required for optimal fiber coupling. While training on the experiment can be difficult for many reasons (see Section 3), it can be the last resort in cases where creating a model that accurately represents the noise and dynamics of the system is very time-consuming, if not infeasible. Using a too-inaccurate model, however, would make it

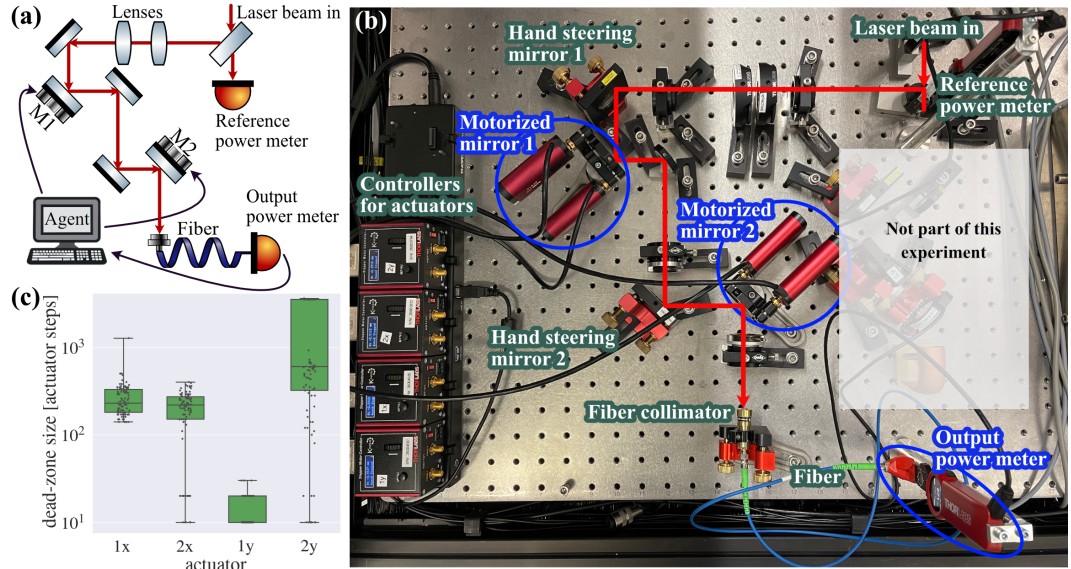

Figure 1: Panel (a) and (b) show a conceptual scheme and lab setup of the fiber coupling experiment. Panel (c) shows the dead-zone characterization of the four motorized mirror mount axes on a log scale axis. Dead-zone means movement steps performed by the actuators that do not result in a change in power. Appendix B gives a detailed description of the characterization.

impossible to cross the reality gap [70]. We therefore decided to study the little-explored field of in-situ training.

## 3 FIBER COUPLING

### 3.1 EXPERIMENTAL SETUP

To efficiently couple laser light into an optical fiber, we need a specific setup. Our goal is to reach a certain coupling efficiency, which is the fraction of light entering the fiber. To achieve this, the light has to enter the fiber at a specific angle and precise spot. The coupling efficiency depends on how accurately both are matched. To fulfill both constraints, we need two degrees of freedom in each axis, horizontal ($x$) and vertical ($y$) [71; 72]. This means that two mirrors, each tiltable in $x$ and $y$, are sufficient to acquire an arbitrary beam alignment. Furthermore, the laser beam must have the correct size, which is achieved by placing lenses in the correct position before the light enters the fiber. To simplify the setup, we decided to motorize only the mirrors but not the lenses. In addition to the motorized mirrors, we have two mirrors that can be steered with hand-tuneable knobs. This makes it easier for humans to couple light into the fiber. We measure the power at the output of the fiber with a power meter (Thorlabs PM160). With the help of a reference measurement, we can determine the coupling efficiency or normalized output power with an error of 2%. Our experiment is depicted in Figure 1, and further details are given in Appendix A.

The four actuators moving the mirrors are stepper motors (Thorlabs ZFS 13). They are attached to the mirror mounts, each tilting the respective mirror in one axis. To understand the special constraints of our problem, we move all actuators to a position where we have maximal coupling. Then, while holding the other three actuators fixed, we scan the relevant movement range with one actuator. The power dependence on each motorized degree of freedom looks Gaussian. Fitting it with a Gaussian, we obtained standard deviations in the range of $10^4 - 2 \times 10^4$ actuator steps.

## 3.2 RL CHALLENGES

When we use RL to fiber couple in our lab, we face several challenges. The training is time-consuming as one actuation step takes about 1 second. Furthermore, due to laser safety and possible equipment damage, we have to restrict the movement range of our actuators. The two challenges that are most crucial for shaping our environment are partial observability and motor imprecision.

**Partial observability** We work with a strongly underdetermined, only partially observable experiment. To describe the state of the experiment perfectly, we would need a lot of information not available to us, e.g., the exact angle, position, and size of the incoming laser beam, as well as the exact position of all of the mirrors and lenses. Even if we could get all of this information at the time of training, environmental drift, such as temperature, would require careful calibration to occur frequently. To make the agent robust against drift, we do not use the actuator positions as part of the observation. Even if we did, they would be very inaccurate due to motor imprecision (see below). Instead, we solely rely on the power at the output of the fiber and the previous actions as our observation. This makes our environment partially observable and underdetermined due to four mirror positions leading to one output. Also, the signal is very scarce, as when the motors leave a certain movement range, no power at all gets coupled into the fiber, which means we do not get any feedback (this is why we reset when falling below a certain power).

**Motor imprecision** Our main challenge is based on the complex relationship between the expected movement of the used motors and their actual movement, which we call *noisy actions*. When we report actuator steps here, these are the steps the controller intended to move the motor. There is no feedback, e.g., an encoder, to ensure the intended position is reached. The imprecision includes backlash of the mechanical system, step loss, non-orthogonal degrees of freedom, i.e., the $x$ and $y$-axis are not independent from each other, and other errors. This leads to noise in the action. To understand its severity, Figure 1 (c) shows the number of steps each actuator moves without any change of power, called *dead-zone*. Although all mirror axes are motorized with the same motor, gearbox, and linear actuator, different dead-zone sizes are observed. A more detailed explanation of the imprecision and its characterization can be found in Appendix B. The variety in the motor imprecision makes the action noise distribution hard to describe. On top of that, this affects our reset method (see Section 4).

## 4 OUR METHOD

We cannot write down a Markov state (see e.g. [5] for an introduction) for our system. Therefore, we treat it as an unknown episodic partially observable Markov decision process (POMDP, see e.g. [73]). Sampling from it, we get a stochastic process $o_1, a_1, r_1, o_2, a_2, r_2, ..., o_\tau$, where $o_t, a_t$ and $r_t$ are observations, actions and rewards at the discretized time $t$, and $\tau$ is the episode length limitited by the maximal episode length $T$, i.e. $\tau \leq T$. See Tables 1 and 2 for environment hyperparameters.

We create a virtual testbed to test out various RL algorithms and investigate differently designed environments before training on the actual experiment. To do so, we fitted the power depending on the position of each individual actuator with Gaussians. By multiplying them, we get a map from all four actuator positions to the power. We then set the amplitude to the highest power we observed until that point, which is 0.92. Using this, we create a simplified virtual environment, not including noise. Although it is a strong simplification, it helped us get various insights much quicker than in the experiment. These numerical results can be found in Appendix C.

**Actions** We treat our action space as the 4-dimensional continuous action space $[-1, 1]^{\times 4}$. At time $t$, we can decompose the action $a_t$ as $a_t = (a_{m1x}, a_{m1y}, a_{m2x}, a_{m2y})$ where each component belongs to a different actuator. For example, $a_{m1x}$ belongs to the actuator that tilts mirror 1 in the horizontal $(x)$ direction. Each of these actions is then multiplied by

the maximum allowed action in actuator steps $a_{\max}$, rounded to the next integer, and sent to the different controllers. Using the virtual testbed, we find that maximum actions of approximately $a_{\max} = 6 \cdot 10^3$ are optimal (see Appendix C.6). However, in the experimental environment, the actuators have no feedback loop, so, potentially, they move significantly less due to their imprecision. This makes our actions noisy, which adds to the stochasticity in the environment. As discussed in Appendix C.8, this especially complicates the task for high powers.

**Observations**   As we are dealing with a partially observable system, for successful training, we need to take great care in defining our observations. The only thing we observe in our environment is the coupling efficiency or normalized power, denoted $P \in [0, 1]$. For example, $P_t$ denotes the normalized power at time $t$. As usual in POMDPs, we include a history of length $n \in \mathbb{N}$ in the observation (see e.g. [66; 61]). This would lead to an observation like $o_t = (P_{t-n}, ..., P_{t-1}, P_t)$ at time step $t$. It is common in RL experiments to observe the environment before and after an action, not during this action. However, this is not the only information available to us: In principle, we can record the power almost continuously while the actuators are moving, i.e., we can record $(P_t, P_{t+1/m_t}, ..., P_{t+1})$ during action $a_t$ where $m_t$ is the number of powers measured during that time. In the virtual testbed, we noticed that it is beneficial to use some of this information in our observation (see Appendix C.5). In particular, we take the average power $P_{\text{ave},t} = \sum_{i=0}^{m_t} P_{t-1+i/m_t}/(m_t + 1)$ the maximal power observed $P_{\max,t} = \max_{i=0,...,m_t} P_{t-1+i/m_t}$ and its relative position in the list of powers $x_{\max,t} = (\arg\max_{i=0,...,m_t} P_{t-1+i/m_t})/(m_t + 1)$ into account. In addition, we want the performed actions to be part of the observation. This leaves us with the observation

$$o_t = \left( (P_{k-1}, a_{k-1}, P_{\text{ave},k}, P_{\max,k}, x_{\max,k})_{k=t-n,...,t}, P_t \right),$$

i.e., the observation includes a history of the power before taking an action, the action, the average and maximum power and its relative position during the action, and the power afterward. Using the virtual testbed, we find that history lengths of approximately $n = 4$ are optimal when using TQC (see Appendix C.5). We deliberately do not take the absolute actuator positions as part of our observation. The main reason is that we want to make our agent robust to experimental alignment changes such as drift. When this happens, the absolute positions where the maximum power is reached will change. An agent trained with the absolute motor positions being part of the observation is not able to handle such situations (see Appendix C.5).

**Episode and Resets**   We reset the environment either after a chosen maximum episode length $t = T \in \mathbb{N}$, when the agent reached its goal (i.e., $P_t > P_{\text{goal}}$), or the agent failed (i.e., $P_t < P_{\text{fail}}$). Using the virtual testbed, we find that it helps with reaching higher goals like $P_{\text{goal}} = 0.9$ if we implement an instance of curriculum learning [74] by starting with lower goal powers and raising the goal power during training, especially starting from $P_{\text{start, goal}} = 0.85$ (see Appendix C.4).
A common way of resetting at the start of an episode is to move to a random position within a defined range. However, our actuators do not present the required precision for this. We, therefore, need a different way to reset. We nevertheless define *neutral positions* given in motor steps. These are positions where we had high power when we started our training. When we return to the neutral positions during training, depending on the original actuator position, the power varies between no power and high power.
The reset procedure depends on the last power value, and we want the power after the reset to be higher than $P_{\min}$. We distinguish between 3 cases. If, during the reset procedure, the

Table 1: Environment parameters for the main experiments where $P_{\min} - 0.1$ is the lowest power where the training starts, $P_{\text{fail}}$ is the power where the agent fails and the reset is called, $P_{\text{goal}}$ is the power the agent should learn to reach, $T$ is the maximal episode length in time steps, and $n$ is the history length used in the observation.

| Parameter | $P_{\min}$ | $P_{\text{fail}}$ | $P_{\text{goal}}$ | $a_{\max}$ | $T$ | $n$ |
|---|---|---|---|---|---|---|
| Value | 0.2 | 0.05 | $[0.8, 0.9]$ | $6 \cdot 10^3$ | 30 | 4 |

Table 2: Reward hyperparameters for the main experiments

| Parameter | $A_s$ | $A_f$ | $A_g$ | $\alpha_s$ | $\alpha_f$ | $\alpha_g$ | $\beta_s$ | $\beta_{f1}$ | $\beta_{f2}$ | $\beta_{g1}$ | $\beta_{g2}$ |
|---|---|---|---|---|---|---|---|---|---|---|---|
| Value | 10 | 100 | 100 | 0.9 | 0.5 | 0.5 | 5 | 5 | 5 | 5 | 1 |

condition of a different case applies, we jump to the corresponding case:

1. $P_t \geq P_{\min}$: we choose a random power between $P_{\min} + 0.1$ and $P_{\text{goal}}$ and do random steps until we decrease the power below the chosen power value.

2. $0.09 < P_t < P_{\min}$: we first reverse the last action. As long as the power is still under $P_{\min}$, we move the actuators one after the other in random order in the direction in which the power increases. If the power decreases, we change the actuator's direction of movement. We repeat this process until we reach $P_t \geq P_{\min}$.

3. $P_t < 0.09$ or every ten episodes: We first move to the neutral positions. From there, we do random steps until $P \geq 0.09$, and then follow the procedure of Case 1 or 2 depending on the power.

The values of 0.09 and ten episodes were determined empirically by observing the algorithm performing on the experiment. Before starting the episode, we always perform some random steps to randomize the process more. Our reset procedure introduces a small dependence between successive episodes. However, this did not affect the training performance in the virtual testbed much (see. Appendix C.3). Furthermore, due to the motors' inaccuracies, full independence was not possible in this experiment.

**Rewards**  We design our reward with the purpose of making the agent reach the goal as quickly as possible. The agent gets a low negative reward when failing ($P_t < P_{\text{fail}}$), a high reward when reaching the goal ($P_t > P_{\text{goal}}$), and else a small reward every step depending on the power. We define the reward function depending on the current power $P_t$, the time step $t$, the goal power $P_{\text{goal}}$, minimal power $P_{\min}$, fail power $P_{\text{fail}}$ and the episode length $T$ as

$$
\begin{aligned}
r_t =\ & R(P_t, t, T, P_{\text{fail}}, P_{\text{goal}}, P_{\min}) \\
=\ & \begin{cases}
-A_f \left( (1 - \alpha_f) \exp\left( -\beta_{f,1} \frac{t}{T} \right) + \alpha_f \exp\left( -\beta_{f,2} \frac{P_t}{P_{\text{fail}}} \right) \right) & \text{if } P_t < P_{\text{fail}} \\
A_g \left( (1 - \alpha_g) \exp\left( -\beta_{g,1} \frac{t}{T} \right) + \alpha_g \exp\left( \beta_{g,2} \frac{P_t}{P_{\text{goal}}} \right) \right) & \text{if } P_t > P_{\text{goal}} \\
\frac{A_s}{T} \left( (1 - \alpha_s) \exp\left( \beta_s (P_t - P_{\text{goal}}) \right) + \alpha_s (P_t - P_{\min}) \right) & \text{else}
\end{cases}
\end{aligned}
$$

where $\beta_s, \beta_{f1}, \beta_{f2}, \beta_{g1}, \beta_{g2}, A_s, A_f, A_g \in \mathbb{R}$, $\alpha_s, \alpha_f, \alpha_g \in (0, 1)$. See Table 2 for their values in the main experiment, which were tuned in the virtual testbed as described in Appendix C.1. The return should never be higher when staying just below the goal than when actually reaching the goal. In the same way, it should always be better to stay just above the failing threshold than to fail. We enforce this by choosing the amplitudes according to $A_f, A_g \geq A_s$. Each of the rewards consists of two terms. The $\alpha$'s are used to weigh their importance relative to one another. The failing reward consists of a term punishing it less when the agent fails later in the episode and a term punishing it less when the power with which the agent fails is close to the failing threshold. The two terms in the goal reward ensure that the agent is rewarded more for reaching the goal quickly and for reaching it with a higher power. The step reward contains both an exponential and a linear part to ensure that the agent clearly notes a change to higher powers for low and high values. As the reward depends on the goal power, we normalize the return in the shown plots by dividing it by the maximum possible return for that given goal power. The maximum possible return is given by the return when reaching the maximum possible power in the first step.

**Algorithms**  The continuous action space limits our choice of algorithm. Of the algorithms tested in the virtual testbed, TQC, TD3, SAC, DDPG, PPO, and Advantage Actor-Critic (A2C), PPO and A2C performed worst. As they both do not use a replay buffer, this was expected. They are closely followed by DDPG. SAC, TD3, and TQC performed much better. SAC performed almost always slightly better than TD3. TQC has a small drop compared to SAC in the middle of training that gets larger with rising goal powers. See Appendix C.7 for a discussion. We used the algorithms from StableBaselines3 [34] with

standard hyperparameters further discussed in Appendix E. In the main experiment, we tested both TQC and SAC as algorithms.

For both the main experiment and virtual testbed, we used a gymnasium environment [75] and the parameters provided in Tables 1 and 2. Our strategy is that the agent learns to deal with the noisy actions directly in the experiment. We are especially interested in investigating this in-situ learning of noise as, in our area, we are often dealing with noise sources that cannot be modeled accurately, for example, when dealing with quantum states of light. In these cases, the only solution will be to learn to handle the noise through direct interaction with the experiment.

## 5 EXPERIMENTAL RESULTS IN THE OPTICS LAB

The agent was trained in our lab on components detailed in Section 3.1. Our training speed is limited by the time the actuators take to move, leading to each environment step taking about a second. We let each training run until its return starts to converge. For $P_{\text{goal}} \leq 0.87$, this took around 20 hours or $4 \cdot 10^4$ steps. That we can train successfully in only $4 \cdot 10^4$ steps is a result of the environment shaping discussed in Appendix C. If we set a very high goal power ($P_{\text{goal}} = 0.9$), the training takes much longer ($2 \cdot 10^5$ time steps, which sums up to nearly 4 days of training).

In the different training runs, we changed algorithms and goal powers. For goal powers between $P_{\text{goal}} = 0.85$ and $P_{\text{goal}} = 0.87$, training runs start to converge at around $4 \cdot 10^4$ steps. For higher goal powers, e.g., $P_{\text{goal}} = 0.9$, this is not the case anymore. This is shown in Figure 2 (a). We tested both SAC and TQC by performing two training runs per algorithm on the experiment with a goal power of $P_{\text{goal}} = 0.85$. We can see that in the beginning, the return rises quicker for SAC but is slightly outperformed by TQC later on. This is why we chose TQC for all other experiments presented here.

When we choose $P_{\text{goal}} = 0.9$, we need significantly more training steps ($\geq 2 \cdot 10^5$). Because of that, as discussed before, we also tested pre-training on lower goals on the experiment, i.e., we started the training with a low goal power $P_{\text{goal}} = 0.85$ and increased it in small increments to $P_{\text{goal}} = 0.9$ over the course of the first $10^5$ training steps and left it at that for the next $10^5$ steps. These training runs are shown in Figure 2 (b). The normalized return of the agent first pre-trained on lower powers always drops after changing to a higher goal power, as it first needs to learn to handle the conditions of the changed environment. We can also see that the normalized return reaches lower values the higher the goal gets. This is expected as the task gets harder each time. The normalized return for the agent pre-trained on lower goals reaches higher values than the one starting from scratch. Additionally, we can use intermediate agents to align the experiment to lower powers. We conclude that curriculum learning was helpful for such a high goal in our experiment (see also [76]). Although we also found pre-training on the virtual testbed with an added noise model helpful (see Appendix D), we focus on in-situ training, as we want to find strategies that can work on experiments where a noise model is hard or impossible to obtain.

To understand the help for our everyday lab work, we tested a few of our agents in fiber coupling (marked with a star in Figure 2 (a) and (b)). The test results are shown in Figure 2 (c)-(e). Each of the RL agents was tested a hundred times. We measure the power over time. If the agent does not reach the goal during an episode, we reset the environment, and the agent tries from there. One episode was at most $40\,\text{s}$, and the longest attempt took around $350\,\text{s}$. Panel (c) shows the power plotted against time for the four tested agents. The agent trained with $P_{\text{goal}} = 0.85$ stays on top of both the agent trained with $P_{\text{goal}} = 0.87$ and one of the agents trained with $P_{\text{goal}} = 0.9$ for some time. This is due to the first agent reaching its goal faster than the other ones. Panel (d) shows the number of steps it took for the same four agents to reach their goal after their last reset, i.e. in the successful episode. As expected, the agent with the lowest goal $P_{\text{goal}} = 0.85$ is the fastest in fiber coupling. Furthermore, of the agents trained with $P_{\text{goal}} = 0.9$, the one pre-trained on lower goal powers reaches this high goal faster. Interestingly, although the agent trained on $P_{\text{goal}} = 0.87$ has to reach a lower goal than the pre-trained one with $P_{\text{goal}} = 0.9$, the former is not faster than the latter. For comparison, we also tested a human expert 25 times on how long they would

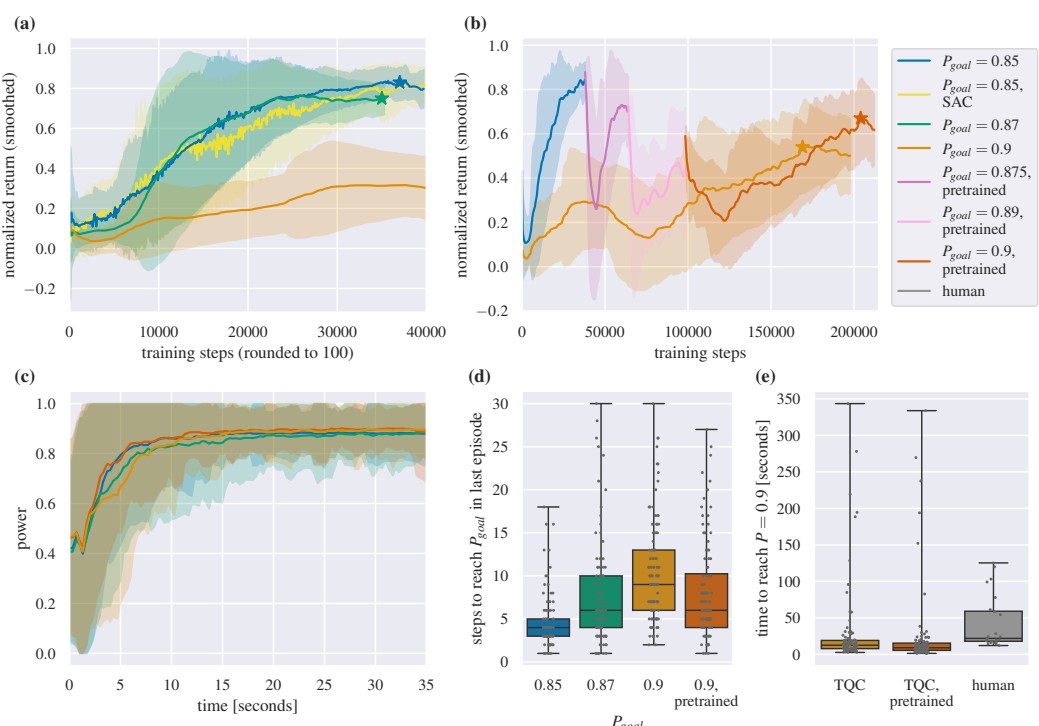

Figure 2: Experimental results. (a) and (b) show the normalized return plotted against training steps for different agents. We use TQC agents except for the yellow curve, where we use SAC. Of the agents trained on $P_{\text{goal}} = 0.9$ (shown in (b))) one agent was first pre-trained on lower, over time increasing, goal powers. (c)-(e) show our results when testing the agents marked with a star in (a) and (b). For testing, we reset and let the agents fiber couple (this is tried 100 times for the RL agent and 25 times for the human expert). If the RL agents do not reach their goal within 30 steps, we reset the environment (still measuring the time), and the agent continues from there. This is repeated until the agent reaches its goal. (c) shows the power plotted against the time for the first 35 s. The error band is clipped to the power range $[0, 1]$. (d) compares the number of environment steps it took for each of the RL agents tested to reach their goal after their last reset. (e) shows the time each agent trained to $P_{\text{goal}} = 0.9$ and a human expert took to reach that goal. Panels (a)-(c) show the (smoothed) mean with $2\sigma$ error bands created by smoothing and/or multiple runs. The error bands in (c) additionally include a power measurement error of 2%.

Table 3: Reset by hand vs. automatic reset. We tested the agent pre-trained on lower goals with $P_{\text{goal}} = 0.9$ as described in Figure 2 (automated reset) or resetting by tilting the hand-steering motors 29 times. We compare the mean of the power at the start $\bar{P}_0$ and end of the episode $\bar{P}_T$, the empirical probability of reaching the goal $p[\text{goal}]$ or failing $p[\text{fail}]$, and mean of the number of environment steps needed to reach the goal $\bar{\tau}_{\text{goal}}$.

|  | $\bar{P}_0$ | $\bar{P}_T$ | $p[\text{goal}]$ | $p[\text{fail}]$ | $\bar{\tau}_{\text{goal}}$ |
|---|---|---|---|---|---|
| Reset by hand | 0.384 | 0.91411 | 0.86 | 0.03 | 8.333 |
| Automatic reset | 0.465 | 0.909936 | 0.9 | 0.03 | 8.22 |

take to couple the fiber to $P_{\text{goal}} = 0.9$ using the hand steering mirrors. This, however, is not a fair comparison. The RL agents can change all four degrees of freedom at once. The experimenter, on the other hand, has access to more information, e.g., the continuous power measurement while moving an actuator, and does not have to deal with the imprecision in the actuators, which means they can easily go back to an observed maximum. Despite this, we can see in Panel (e) that the RL agents are generally faster but take longer in a few cases, where the agent needs several episodes to get to the goal. Our hypothesis is that this is due to our episodes not being fully independent of each other. In conclusion, we show that by using RL, we can consistently couple light into an optical fiber to high efficiencies, despite the noisy actions.

So far, we have shown that our agent can couple light into the fiber after a reset using the motors that it has access to but not after a general misalignment or drift in the setup. To show that the agent can also compensate for misalignment in other parts of the experiment, we performed the resets by manually misaligning the hand steering mirrors, e.g., tilting them until we were in a coupling regime with low power. Next, we called the agent for realignment. Table 3 shows that the results using this alternative reset method are very similar to the ones with automatic reset. Whether such a misalignment happened at the hand steering mirrors or another element not accessible to the agent, such as, for example, a drift in the position of the fiber collimator, is equivalent in terms of difficulty. Hence, our agent can also be used to control for arbitrary drifts at an undetermined location. For this, we can use the almost continuous measurement of the power at the output of the fiber and call the agent to set it back to the desired coupling efficiency whenever it drops below a certain value.

## 6 SUMMARY AND OUTLOOK

We have shown that our model-free RL agent successfully learns to couple laser light into an optical fiber, reaching the same efficiencies as a human expert while generally being faster. We find that sample-efficient algorithms that use a replay buffer, such as SAC and TQC, are a must to overcome the challenge of otherwise not manageable training time. Partial observability can be dealt with by carefully tuning the observations. Furthermore, our study suggests that curriculum learning can help to achieve more difficult goals.

As we train directly on the experiment, the agent learns to handle the specific noise present, and we can avoid creating an accurate simulation of the task. This makes our method suitable for setups where it is impossible to model the noise accurately.

A central result is that the agent learns to deal with the imprecision of the mirror steering motors. Using a classic algorithm such as gradient descent would fail with these motors as their imprecision deters us from experimentally determining a gradient and then going back to the starting position. One way to handle such imprecision is using motors with internal feedback loops. Using RL, we can avoid this, which helps to simplify the design of motors and experimental setups. Generally, automation gives us the possibility of remote-controlling experiments, which can be especially useful in experimental areas that are difficult to access, such as in vacuum tanks, in clean room facilities, or, in extreme cases, in underground labs or in space.

In our experiment, we used four motors to cover all degrees of freedom and demonstrated the general case of laser beam alignment. Reducing the number of actuators per axis to 1

lowers the possible coupling efficiency. In contrast, increasing the number of mirror actuators offers no physical advantage. More complex setups can be divided into parts with multiple mirror-mirror-sensor blocks.

Further exploration could include investigations on how the agents perform for other transverse optical modes of light, including multiple local maxima, and investigating the performance of model-based or hybrid algorithms. We start our RL training procedure under conditions where there is low power. This raises the question how, or with how many additional sensors, we could generalize this to the case of starting with no power. Furthermore, it might be interesting to explore replacing the use of a history as our observation by PID-inspired RL [77] and to investigate whether pretraining on expert demonstrations could speed up the learning process [78–80].

Our experiments are a first step towards the extensive use of RL in our quantum optics laboratory. Optical experiments typically require various control loops to stabilize the experimental degrees of freedom against perturbations. These locks significantly increase the complexity of the experiments. For example, maintaining the length of optical cavities to achieve resonant light field enhancement requires complex components such as phase modulators [81], homodyne detectors [82; 83], or split detectors [84; 85]. RL offers the potential for streamlined control loops that rely solely on the measurement of power in the reflection and transmission of the resonator. This could enable novel control strategies, such as phase control of squeezed vacuum states. These states are characterized by unique quantum noise properties but are otherwise dark. Consequently, phase control typically requires an auxiliary laser field [86] or introduces unwanted phase noise [87]. RL has the potential to provide a noise-free solution without the need for additional laser fields, which is particularly relevant for large-scale on-chip squeezing experiments in the field of quantum information [88].

In conclusion, we show that a common optics task like beam alignment can be solved with standard model-free RL algorithms. For the machine learning community, this demonstrates their versatility. Their availability lowers the barrier for the optics community to use them in other experiments. We showcase how these RL algorithms can be directly applied in the lab, circumventing the need for accurate experimental modeling.

**Reproducibility**   We provide schematics and a detailed description of our experimental setup (Section 3 and Appendix A), which can be used to rebuild the experiment. We also describe how we create our virtual testbed (Appendix C). Furthermore, we explain in detail how we implement the RL algorithms and their hyperparameters (Appendix E). Additionally, all code used for this project and data from the experiment (return and test results) are provided in the supplementary material.

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

## A  ADDITIONAL DETAILS ABOUT THE EXPERIMENTAL SETUP

Our setup includes the following components: We use a 1064 nm laser (Mephisto, Coherent), a single mode polarization-maintaining fiber, and a Schäfter+Kirchhoff fiber collimator (60FC-SF-4-M8-08) at the input side of the fiber. The measurements of laser power are done with power meters (Thorlabs PM160, measurement error 1%). In front of the experiment, we place a partially reflecting mirror to measure a fraction of the laser light with an additional power meter for power reference. The measurement is used to pause training in the event of a laser failure and to track power fluctuations to determine the maximum power level. In this way, we can determine the coupling efficiency with an error of 2%. The input power is set to $1.00(1)$ mW.

For training on the experiment, we used an NVIDIA GeForce RTX 4070 GPU. In addition to the usual packages [89–92], we used PyLabLib, Thorlabs Kinesis, PyVisa, Keysight Connection Expert and safe-exit [93–97] for communication with the experiment.

Due to safety constraints, we have to limit our state space and, in consequence, clip our actions if the actuator positions would otherwise move out of a certain range because it is unacceptable for the laser beam to wander around the room. Also, in the first test run, the action size was chosen so poorly that the mirror mounts were damaged. So, both laser safety and equipment damage are hazards that we need to consider.

## B  EXPLANATION OF THE IMPRECISION IN THE MIRROR STEERING MOTORS AND ITS CHARACTERIZATION

Backlash is a phenomenon that is present when a load is not directly connected to a motor, such as in geared mechanical systems [98]. Dependent on the exact geometry of the system, i.e., mechanical tolerances, amount of gears, etc., it may resemble hysteresis between the expected and actual position or a *dead-zone*, where moving the actuator has no effect on the actual position whenever the rotational direction is reversed. It is thus hard to model and predict a priori. Control has to be implemented based on the specific system and its use. These control schemes include hysteresis models, dead-zone models, and PI control. However, additional sensors are needed to get accurate feedback if multiple actuators are used. Step loss results from the difference in static and dynamic torque of a motor. The motor steps result in a linear actuation, which changes the tilt of the mirror mount. Different tilt angles lead to different static loading of the motor and gearbox. This may lead to the initial step(s) being lost, as the motor can not deliver the starting torque, resulting in a partial step. Without feedback from, e.g., an encoder, this leads to a difference between the expected and actual position. Lastly, the non-orthogonal degrees of freedom are a result of the kinematic mirror mounts used and their mounting. Usually, this error is small for well-designed kinematic mirror mounts.

We performed a dead-zone characterization. The core idea is to initiate a number of movements, i.e., generate a movement history, after which a maximum dead-zone is expected. This can simply be an initial long movement in one direction followed by a direction reverse. The long movement ensures a nearly linear behavior between the expected and actual position, as backlash is overcome in the mechanical system. The backlash after a change in rotational direction should, therefore, be large. Additionally, the movement history is similar between repetitions, enabling their comparison. Starting from a position with high coupling, we moved one actuator far out, then back to high coupling. From there, we reversed the movement and counted the steps the actuator had to move before the measured power changed by more than the power measurement error. Repeating this process 100 times yielded the distribution of the dead-zone size, shown in Figure 1 (c) for the four motors. This data helps us understand the uncertainty of the mirror mount movements. As no continuous feedback is employed, characterization of other positioning errors is not possible in our setup.

## C  Environment and agent tuning on virtual testbed

We used the data of scanning each motor individually through the coupling peak to create a virtual testbed. Each dataset was normalized and fitted with a Gaussian; all of them were then multiplied. The highest coupling efficiency we had measured up to this point was 0.92; therefore, we use this as the amplitude. The following function, based on the fit values in motor steps, describes our virtual testbed:

$$P(x_{m1}, y_{m1}, x_{m2}, y_{m2}) = 0.92\exp\left(-\frac{1}{2}\left(\left(\frac{x_{m1} - 5470785}{11994}\right)^2 + \left(\frac{y_{m1} - 5573194}{19145}\right)^2\right.\right.$$

$$\left.\left. + \left(\frac{x_{m2} - 5461786}{12769}\right)^2 + \left(\frac{y_{m2} - 5178016}{17885}\right)^2\right)\right)$$

We use this testbed to gain insights into the environment hyperparameters, observations, and algorithms to use in the following order.

First, we optimized the hyperparameters of the reward ($\alpha$'s, $\beta$'s, $A$'s). Next, we went to the parameters of the environment that appear in the reward, i.e., the goal power $P_{\text{goal}}$ and episode length $T$. The usual figure of merit is the normalized return in dependence on the training step. However, this depends on the reward, and the reward depends on both of these sets of parameters. Therefore, it is not possible to use the return as a figure of merit for these parameters. Instead, we trained a TQC agent for a total of $10^5$ timesteps. We tested the agent every $10^4$ timesteps for 100 episodes, noting the probability of failure, the probability of reaching the goal, and the average power at the end of each episode. Our main figure of merit was the probability of reaching the goal after a training time in the range of $10^4$ to $4 \cdot 10^4$ time steps. Still, we also took the probability of failure and the average power at the end of each episode into account. We show the second one here only when we used it for our decision.

After fixing the first two sets of parameters, we were able to use the normalized return to compare other environment parameters, such as the length of the history in the observation and the maximal action, and different algorithms. All studies in the virtual testbed were performed at least 5 times and, except for the algorithm tests, used TQC as the algorithm as this was the algorithm most used in the experiment. If not stated otherwise and if these were not the parameters being changed, we used the parameters in Tables 1 and 2.

### C.1  Reward Hyperparameters

We want a high probability of reaching the goal after the least amount of training time, so we shape the reward function accordingly. For tuning its hyperparameters, we chose $P_{\text{fail}} = 0.2$, $P_{\text{min}} = 0.4$, $P_{\text{goal}} = 0.8$, $T = 20$, $\alpha_s = 0.5$, $\alpha_f = 0.9$, $A_f = 10$, in contrast to Tables 1 and 2, if those parameters were not the ones being changed. We tested the tuples of reward parameters given in Table 4. For each parameter we tested a number of different values and also checked the dependence of the variables on each other. After evaluation, we decided on the parameters in Table 2. The subscript $s$ always refers to the step reward, $f$ to the fail reward, and $g$ to the goal reward. The other parameters were $P_{\text{fail}} = 0.2$, $P_{\text{min}} = 0.4$, $P_{\text{goal}} = 0.8$, $T = 20$, $\alpha_s = 0.5$, $\alpha_f = 0.9$, $A_f = 10$ or given in Tables 1 and 2.

Table 4: test table tuning parameters

| Parameter | $(A_f, A_g)$ | $(\alpha_s, \beta_s)$ | $(\alpha_f, \beta_{f1}, \beta_{f2})$ | $(\alpha_g, \beta_{g1}, \beta_{g2})$ |
|---|---|---|---|---|
| Value | $\{10, 100, 1000\}^2$ | $\{0.1, 0.5, 0.9\}$ $\times\{1, 5, 10\}$ | $\{0.1, 0.5, 0.9\}$ $\times\{1, 5\}^2$ | $\{0.1, 0.5, 0.9\}$ $\times\{1, 5\}^2$ |

**Prefactors**  First, we tested different prefactors $A_f$ and $A_g$. The results are shown in Figure 3. Looking at the probability of reaching the goal, $A_f = 100$ seems to be the best value. For training steps over $3 \cdot 10^4$, we can see that $A_g = 1000$ seems to be better than the other two values, before it seems that $A_g = 100$ is doing better. However, if we look at the

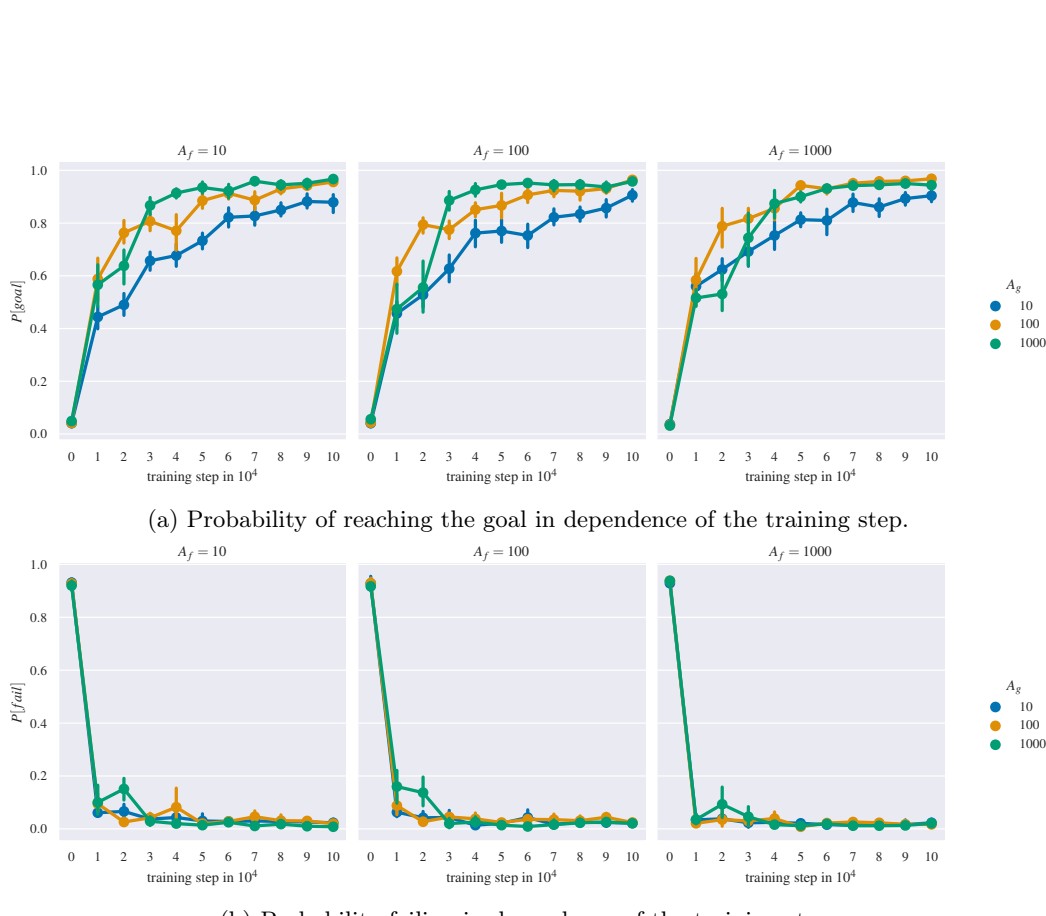

(a) Probability of reaching the goal in dependence of the training step.

(b) Probability failing in dependence of the training step.

Figure 3: Results for prefactor tuning: Probability of reaching the goal or failing for different prefactors in the reward using TQC and $P_{\text{fail}} = 0.2$, $P_{\text{min}} = 0.4$, $P_{\text{goal}} = 0.8$, $T = 20$, $\alpha_s = 0.5$, $\alpha_f = 0.9$ and otherwise the parameters in Tables 1 and 2. The plots show the mean with $2\sigma$ error bars created by multiple runs.

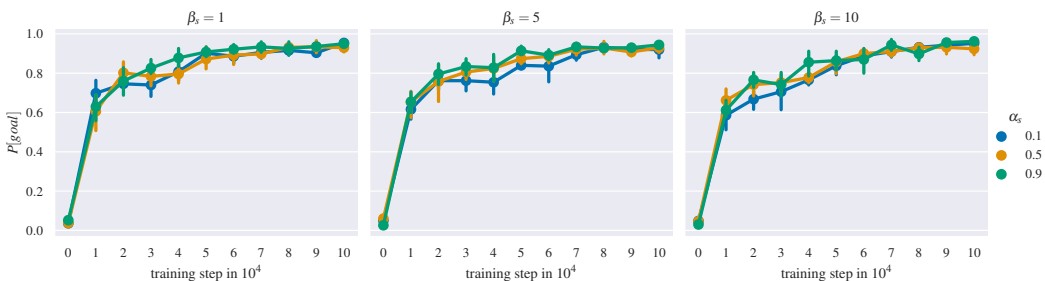

Figure 4: Results for tuning the parameters of the step reward: Probability of reaching the goal for different $\alpha_s, \beta_s$ in the reward with TQC, $P_{\text{fail}} = 0.2$, $P_{\min} = 0.4$, $P_{\text{goal}} = 0.8$, $T = 20$, $\alpha_f = 0.9$, $A_f = 10$ and all other parameters as in Tables 1 and 2. The plots show the mean with $2\sigma$ error bars created by multiple runs.

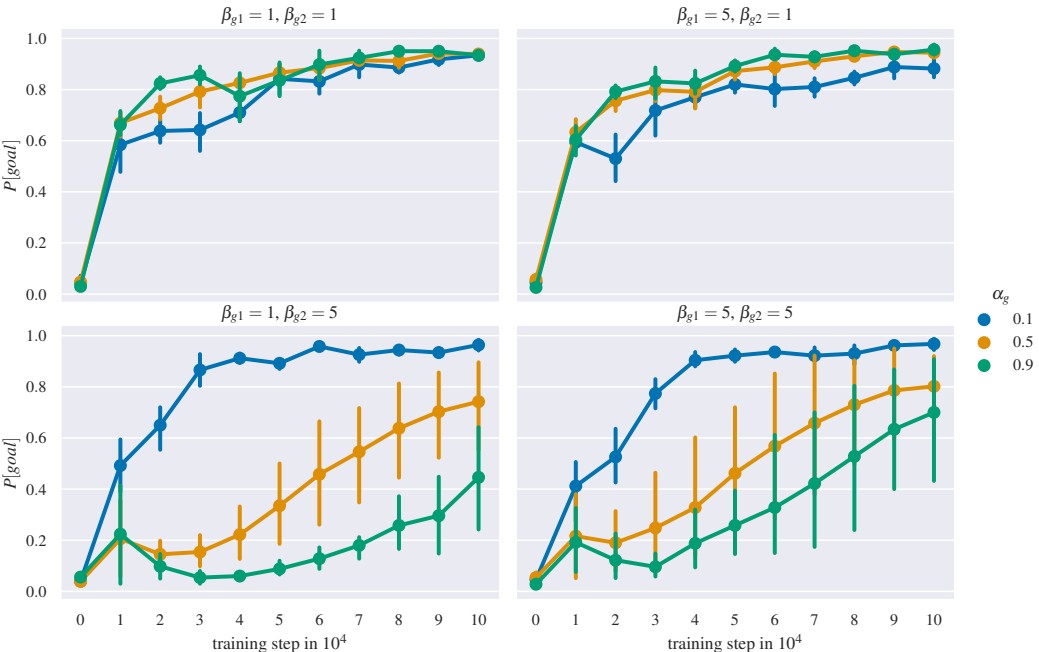

Figure 5: Results for tuning the parameters of the goal reward: Probability of reaching the goal for different $\alpha_g, \beta_{g1}, \beta_{g2}$ in the reward with TQC, $P_{\text{fail}} = 0.2$, $P_{\min} = 0.4$, $P_{\text{goal}} = 0.8$, $T = 20$, $\alpha_s = 0.5$, $\alpha_f = 0.9$, $A_f = 10$ and all other parameters as in Tables 1 and 2. The plots show the mean with $2\sigma$ error bars created by multiple runs.

probability of failure, we can see that using $A_g = 100$, the probability of failure falls more quickly than if we are using $A_g = 1000$. Because resets after failure take a lot of time for this kind of $P_{\min}$ and $P_{\text{fail}}$, we want the failure probability to be as low as possible and go with $A_g = 100$.

**Step Reward** Second, we are looking at the step reward and optimizing for $\alpha_s$ and $\beta_s$. Figure 4 shows the probability of reaching the goal. We deem $\beta_s = 5$ and $\alpha_s = 0.9$ to be the best parameters, although there is not a very strong difference.

**Goal Reward** Third, we are looking at the goal reward and optimizing for $\alpha_g, \beta_{g1}$ and $\beta_{g2}$. Figure 5 shows the probability of reaching the goal. Here, there is a stronger difference between the different parameters. We are going with $\alpha_g = 0.5, \beta_{g1} = 5$ and $\beta_{g2} = 1$. The other possibility would be $\alpha_g = 0.1, \beta_{g1} = 1$ and $\beta_{g2} = 5$, which is worse in training steps

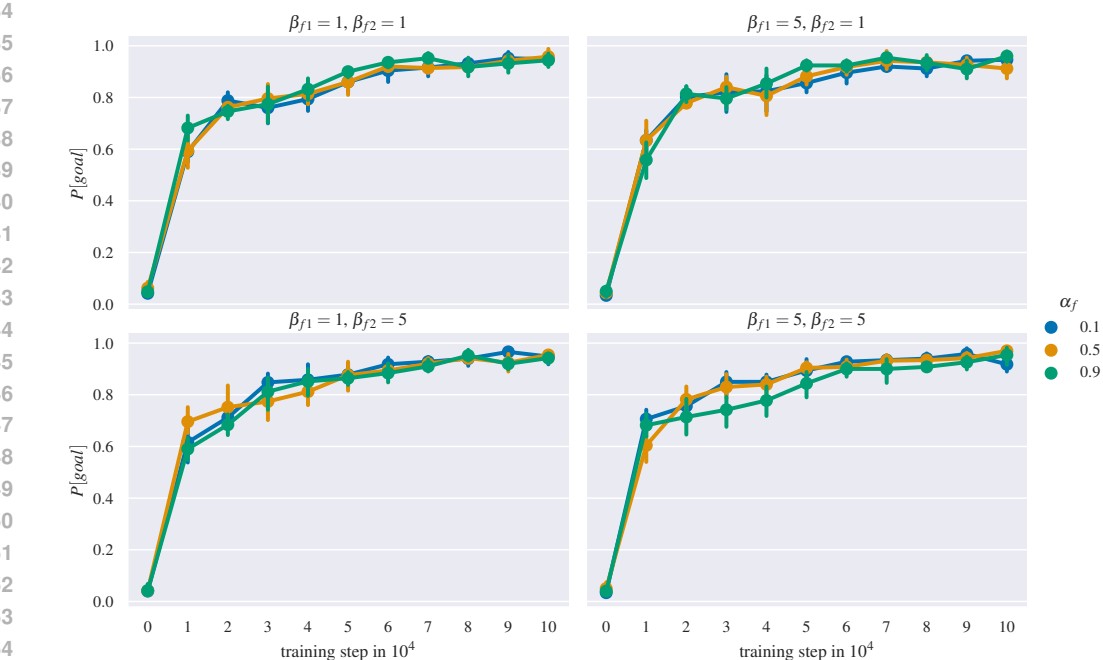

Figure 6: Results for tuning the parameters of the fail reward: Probability of reaching the goal for different $\alpha_f, \beta_{f1}, \beta_{f2}$ in the reward with TQC, $P_{\text{fail}} = 0.2$, $P_{\min} = 0.4$, $P_{\text{goal}} = 0.8$, $T = 20$, $\alpha_s = 0.5$, $A_f = 10$ and all other parameters as in Tables 1 and 2. The plots show the mean with $2\sigma$ error bars created by multiple runs.

$1 \cdot 10^4 - 2 \cdot 10^4$ but better in training steps $4 \cdot 10^4 - 5 \cdot 10^4$. However, we put our focus on the earlier phases of training and also do not want to emphasize the power with which the goal was reached that much over the time in which it was reached, which is why we go with the first choice of parameters.

**Fail Reward** Lastly, we are looking at the fail reward and optimizing for $\alpha_f, \beta_{f1}$ and $\beta_{f2}$. Figure 6 shows the probability of reaching the goal. Here, the choice is again not that clear, but we deem $\beta_{f1} = \beta_{f2} = 5$ and $\alpha_f = 0.5$ to be the best choice of parameters.

## C.2 Episode Length

We want to find a good trade-off between reaching the goal quickly and reaching it reliably. Using the same parameters as for reward shaping, we tested different episode lengths, in particular, $T = 5, 10, ..., 50$. The results are shown in Figure 7. As expected, the longer the episode, the higher the probability of reaching the goal (or failing). For some of these (i. e. $T = 20, 30, 35, 40$, we also varied the maximum allowed actuator steps per environment step $a_{\max}$ (i. e. doing simulations with $a_{\max} = [2 \cdot 10^3, 10^4]$ to see if it had an effect on this, which we could not confirm. However, we also have to take into account that longer episodes will take more time in the experiment. This is why we select $T = 30$, as there is not a very big difference between this and $T > 30$.

## C.3 Reset Methods

In the virtual testbed, we compare the following reset methods:

    A Reset as described in the main paper (for testing at the start and end of training).

    B Reset as described in the main paper, but first, go to neutral positions in every episode.

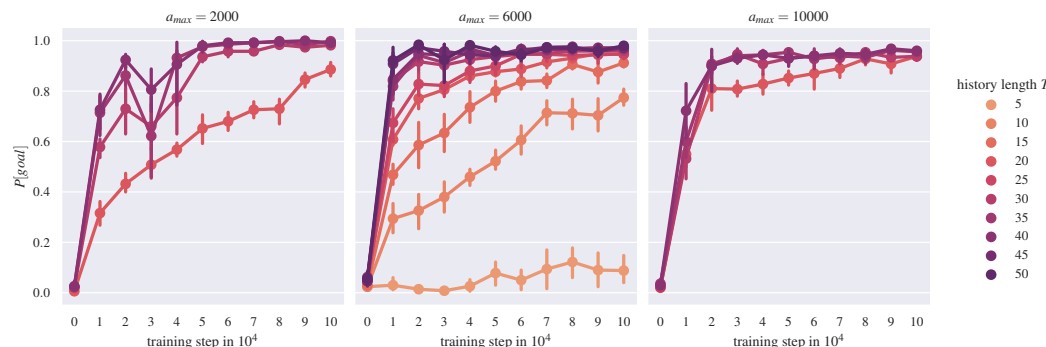

Figure 7: The probability of reaching the goal in dependence of the training step for different maximum episode lengths $T$ and maximum actions $a_{max}$ with TQC, $P_{\text{fail}} = 0.2$, $P_{\text{min}} = 0.4$, $P_{\text{goal}} = 0.8$, $T = 20$, $\alpha_s = 0.5$, $\alpha_f = 0.9$, $A_f = 10$ and all other parameters as in Tables 1 and 2. The plots show the mean with $2\sigma$ error bars created by multiple runs.

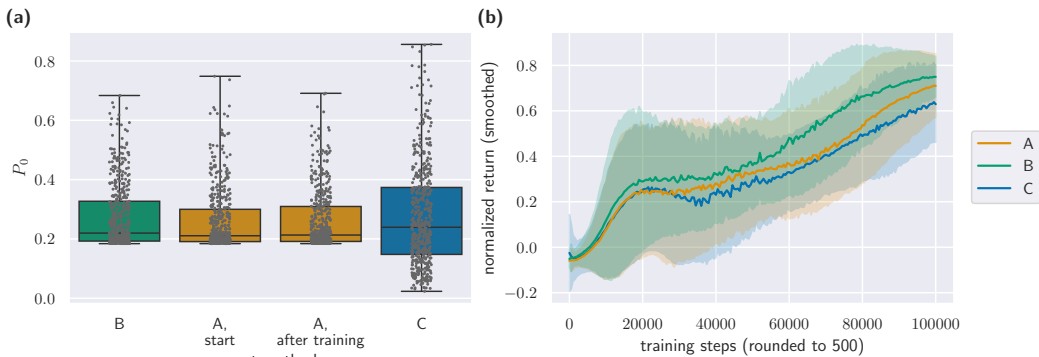

Figure 8: Comparison of different reset methods. The methods are the following: A – Reset as described in the main paper (for testing at the start and end of training), B – Reset as described in the main paper, but first, go to neutral positions in every episode, C – Reset by choosing random positions for all four actuators in an interval of width $4.2 \cdot 10^4$ around the neutral positions. We use the parameters from Tables 1 and 2, TQC and $P_{\text{goal}} = 0.85$. (a) shows a box plot of the starting powers. (b) shows the normalized return in dependence on time. The mean is shown with $2\sigma$ error bands created by smoothing and multiple runs.

> C Reset by choosing random positions for all four actuators in an interval of width $4.2 \cdot 10^4$ around the neutral positions.

We used the parameters from Tables 1 and 2 and $P_{\text{goal}} = 0.85$. The results can be seen in Figure 8. In Panel (a), we can see the starting power for the different reset methods. Using method C, the starting distribution of powers is very different from the other reset methods. The median is similar, but the standard deviation is much higher. This led us to compare methods A and B additionaly. In method B, the median is slightly higher and independent of our policy. For method A, the distribution depends on the model used, and the median and 75$^{\text{th}}$ quantile are slightly higher and more comparable to the one of B after $10^5$ training steps than at the start. Because of this, the return for method B is slightly higher than for method A, especially in the middle, as can be seen in Panel (b). We would have expected a higher impact from the reset method. However, the differences are quite small. In conclusion, even though our method makes our episodes not fully independent of each other, we do not gain an artificial benefit from it.

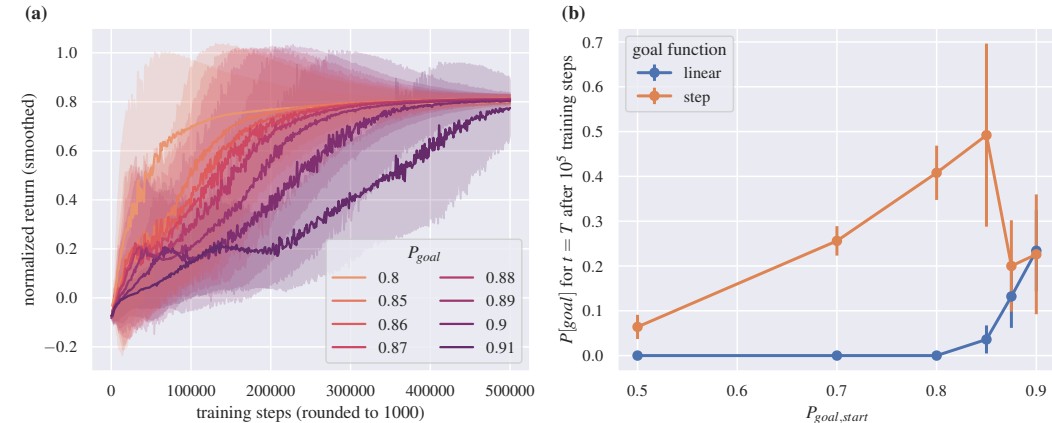

Figure 9: (a) shows the normalized return in dependence of the training step for different goal powers. (b) shows the probability of reaching the goal power $P_{\text{goal}} = 0.9$ after $10^5$ training steps in dependence of $P_{\text{goal, start}}$. Hereby, the goal on which the model is trained either rises in a linear (orange) or step (blue) function of the training step from $P_{\text{goal, start}}$ to $P_{\text{goal}} = 0.9$ over the course of $10^5$ training steps. Both panels show the mean with $2\sigma$ error bands created by multiple runs and, in (a), smoothing.

## C.4 GOAL

The goal power is fully our choice. First, we check at what point the return starts to converge for which goal power. Therefore, we look at the normalized return in dependence on the training steps for different goal powers. This is shown in Figure 9 (a). We can see that the higher the goal power, the lower the normalized return after convergence, and the later the return converges. We can see that this point happens significantly later for high goal powers. Also, for high goal powers like $P_{\text{goal}} = 0.9$, the distance to the last curve is bigger than, for example, for $P_{\text{goal}} = 0.86$.

Because of this, we wondered if it made sense to pre-train on lower goal powers. We tested this by starting with goal powers $P_{\text{start, goal}} = 0.5, 0.7, 0.8, 0.85, 0.875$ and raising it to 0.9 over the course of $10^5$ training steps either linearly, i.e., raising it slightly in every training step, or in a staircase way, i.e. raising it more every $10^4$ training steps. Figure 9 shows the probability of reaching the goal $P_{\text{goal}} = 0.9$ after $10^5$ training steps in dependence of the starting goal power $P_{\text{start, goal}}$ for the two different manners of raising the goal power. We can see that it can be helpful to raise the goal power in steps, especially starting from $P_{\text{start, goal}} = 0.85$.

## C.5 OBSERVATION

We tested history lengths of $n = 1, ..., 6$ and the maximum sensible length $n = T = 30$ with $P_{\text{goal}} = 0.85$. The results can be found in Figure 10 (a). Depending on the training step, $n = 3, 4$ lead to the highest return. We went with $n = 4$ as this was higher around $2 \cdot 10^4$ to $5 \cdot 10^4$ training steps.

We tested if removing $P_{\text{ave}}$ or $P_{\text{max}}$ and $x_{\text{max}}$ from the observation influences the performance. Figure 10 (b) shows that TQC performed worse on any of these combinations compared to the full observation presented above. However, leaving out $P_{\text{ave}}$ had a much smaller impact than leaving out $P_{\text{max}}$ and $x_{\text{max}}$, which makes the latter very important for us.

Additionally, we tested how agents perform in an environment that includes the absolute position of the actuators in the observation compared to one that does not. The normalized return against the training step can be found for both configurations, using the parameters in Tables 1 and 2, TQC, and $P_{\text{goal}} = 0.85$, in Figure 10 (c). As expected, the agent learns faster and reaches a higher normalized return if those absolute positions are included. However,

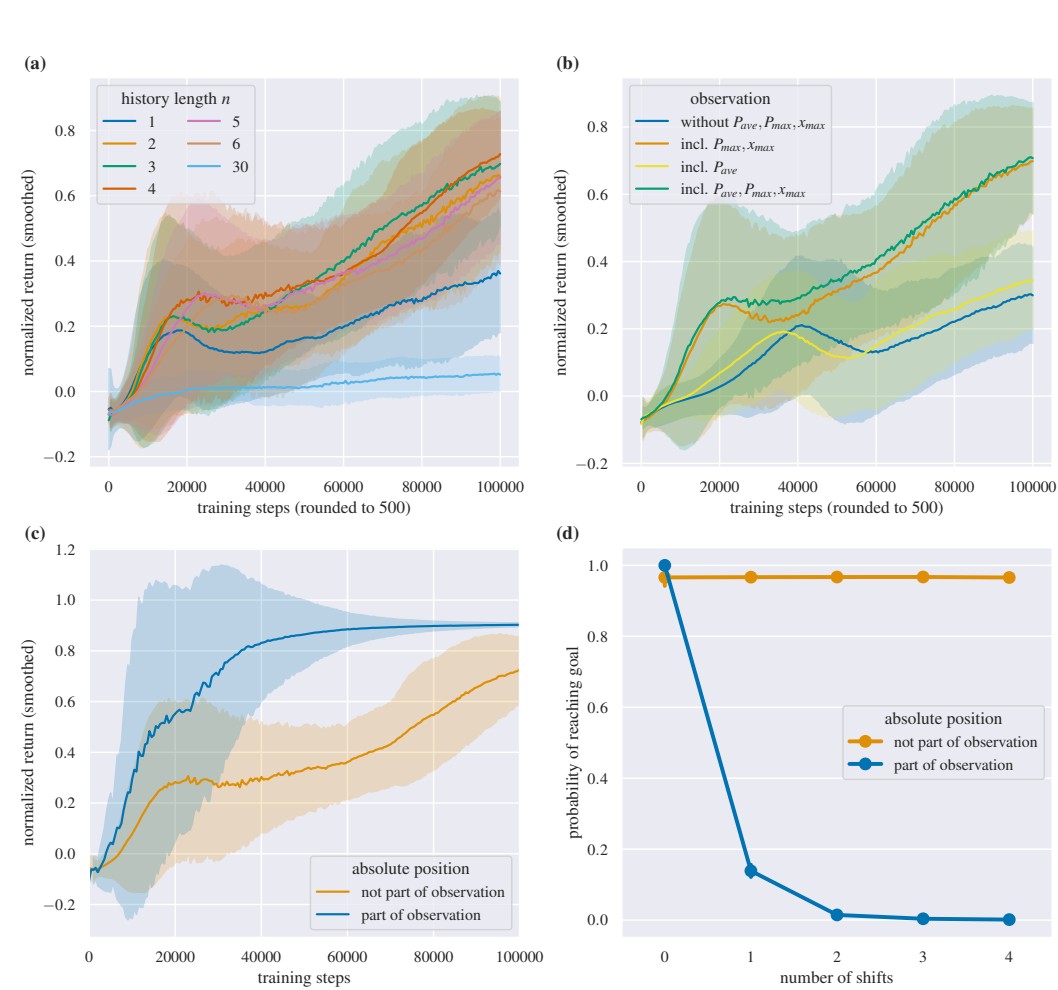

Figure 10: Comparison of different observations and maximum actions. (a)-(c) show the normalized return in dependence of the training step using the parameters in Tables 1 and 2, TQC, and $P_{\text{goal}} = 0.85$ for different history lengths $n$ in (a), leaving out different parts of the observation in (b), and with or without including the absolute positions in the observation in (c). The models with and without the absolute positions were then tested 100 times each in an environment in which $k = 0, ..., 4$ means of the underlying Gaussian $\mu_i$ were shifted by $\pm \sigma_i$, i.e., $\mu'_i = \mu_i \pm \sigma_i$. (d) shows the probability of reaching the goal against the number of shifts $k$. The plots show the mean with $2\sigma$ error bands/bars created by multiple runs and, in (a)-(c), smoothing.

the main application for our agent is to recouple the light into the fiber after other parts of the experiment have been misaligned. That means, that the optimal positions, i.e., the means $\mu_i$ of the underlying Gaussian, change, and the agent still has to be able to reach the goal. Hence, we tested the agent 100 times each in environments in which $k = 0, ..., 4$ means of the underlying Gaussian $\mu_i$ were shifted by $\pm\sigma_i$, i.e., $\mu'_i = \mu_i \pm \sigma_i$. Figure 10 (d) shows the probability of reaching the goal against $k$. If no shifts occur, the agent with the absolute position as part of the observation performs slightly better than the one without. However, this quickly changes as more shifts are applied. The agent with absolute position performs much worse if any shifts appear. On the other hand, the agent not observing the absolute position performs well independent of shifts.

### C.6 Action

We tested different maximal actions $a_{\max} = 2 \cdot 10^3, 4 \cdot 10^3, 5 \cdot 10^3, ..., 10^4$ with $P_{\text{goal}} = 0.85$, TQC, and the other parameters as in Tables 1 and 2 to see which yields the highest return. The results are shown in Figure 11 (a). Maximum actions between $4 \cdot 10^3$ and $8 \cdot 10^3$ generally performed best ($4 \cdot 10^3$ performed best out of them). Because of the imprecision of the motors, we also did some tests in the lab, which is why we selected $a_{\max} = 6 \cdot 10^3$ for our experiments. This is approximately half of the standard deviation of the Gaussian in $x-$ direction.

### C.7 Algorithms

We tested six different algorithms with their standard hyperparameters in StableBaselines3 with the parameters in Tables 1 and 2 for $P_{\text{goal}} = 0.8, 0.85, 0.9$ for either $10^5$ (for $P_{\text{goal}} = 0.8, 0.85$) steps or $5 \cdot 10^5$ (for $P_{\text{goal}} = 0.9$) training steps. The results are shown in Figure 11 Panel (b)-(d). We can see that in the first $10^5$ steps, A2C and PPO always perform worst, and DDPG is next in line. However in Figure 11 (d), we can see that for $P_{\text{goal}} = 0.9$ PPO catches up to DDPG around training step $1.5 \cdot 10^5$. SAC, TQC, and TD3 perform much better than these three. TD3 is nearly always worse than SAC. TQC always has a drop in the middle region but catches up to SAC in the end. Overall, SAC seems to be the best algorithm for this task when used on the virtual testbed. In contrast to that, in our physical experiments, TQC slightly outperforms SAC for $P_{\text{goal}} = 0.85$.

### C.8 Effect of noise on the learning process

We use the characterization of the dead-zone in the actuators to derive a simple noise model: Each time the agent performs an action, its size is reduced by a value randomly sampled from the dead-zone characterization multiplied by a noise factor $\mathcal{N}$. For a noise factor of $\mathcal{N} = 0$, there is no noise, and the results are similar to the ones discussed in the virtual testbed section up to this point. A noise factor of $\mathcal{N} = 1$ should make the noise level of the virtual testbed comparable to the one present in the experiment. In comparison, noise factors of $\mathcal{N} > 2$ correspond to higher noise levels than in the experiment. For $P_{\text{goal}} = 0.85$, 0.9 and noise factors of $\mathcal{N} = 0, ..., 3$, the normalized return is shown in Figure 12. As expected, the return for higher noise levels is generally smaller than the one for no noise, but for each of the presented noise levels, the agents are still able to learn. For $P_{\text{goal}} = 0.85$, the graphs for $\mathcal{N} = 0$, 1 are very similar and only for the higher noise factors ($\mathcal{N} = 2$, 3) the learning curve clearly differs. That means that for a moderate goal power, the noise in the experiment does not affect the agent as much. However, this is different for $P_{\text{goal}} = 0.9$, the graph for $\mathcal{N} = 1$ is grouped with the ones for $\mathcal{N} = 2$, 3. Hence, the agents' learning curves are significantly impacted by the noise level found in the experiment.

## D Other experimental runs in the optics lab

### D.1 Different goal powers

We run experiments with $P_{\text{goal}} = 0.85, 0.86, 0.87, 0.88, 0.9$ using TQC. The normalized return is shown in Figure 13 (a). We can see that, just like in the virtual testbed, the training

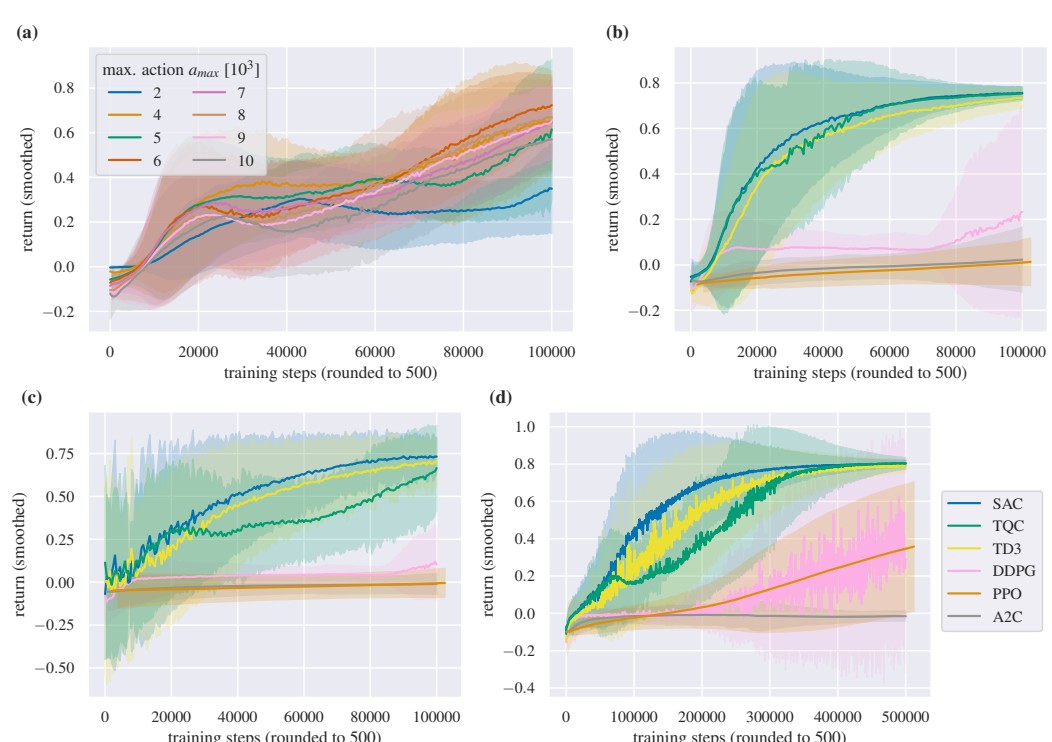

Figure 11: Comparison of different maximum actions and algorithms. The plots show the normalized return against the training step for different maximum actions $a_{\mathrm{max}}$ and TQC in (a) or for six different algorithms: TQC, SAC, TD3, PPO, DDPG and A2C in (b)-(d). (b) shows this for $P_{\mathrm{goal}} = 0.8$, (c) for $P_{\mathrm{goal}} = 0.85$, and (b) for $P_{\mathrm{goal}} = 0.9$. The other parameters are chosen as in Tables 1 and 2. The plots show the mean with $2\sigma$ error bands created by multiple runs and smoothing.

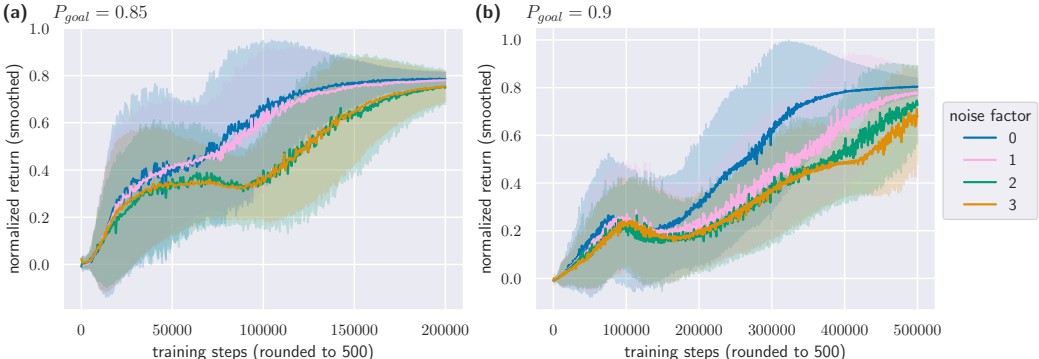

Figure 12: Comparison of different noise levels. The plots show the normalized return against the training step for different goal powers $P_{\mathrm{goal}} = 0.85$, 0.9 and noise factors $\mathcal{N} = 0, ..., 3$ using TQC. The other parameters are chosen as in Tables 1 and 2. Both panels show the mean with $2\sigma$ error bands created by multiple runs and smoothing.

needs longer to converge the higher the goal. It is interesting that there is a big gap between the agents with $P_{\text{goal}} = 0.87$ and $P_{\text{goal}} = 0.88$. Please note that we performed these training runs (except for $P_{\text{goal}} = 0.85$ only once and draw our conclusions from there.

## D.2 Replay buffer

We already discussed in the main paper that it can make sense to pre-train agents on lower goals. However, we did not discuss what we do with the replay buffer when changing the goal power. Here, we test if it would be better to keep or delete it when changing to the next higher goal power. We perform two training runs on the experiment starting with $P_{\text{goal}} = 0.85$, raising it to $P_{\text{goal}} = 0.875$ after $3.8 \cdot 10^4$ training steps, then raising it to $P_{\text{goal}} = 0.89$ after approximately $6.35 \cdot 10^4$ training steps, and then raising it to $P_{\text{goal}} = 0.9$ after $9.8 \cdot 10^4$ training steps. In the first, we delete the replay buffer after changing our goal to $P_{\text{goal}} = 0.875$ and $P_{\text{goal}} = 0.89$ (yellow, discussed in main paper); in the second, we do not (green). Both runs are shown in Figure 13 (b). In the yellow ones we see more drops after each rise in goal power, but its normalized return is slightly higher in the end. Overall, the results are quite similar.

## D.3 Pre-training on virtual testbed

Furthermore, we want to know if pretraining on the virtual testbed helps with the experiment's training times. We tested both an agent pre-trained on the virtual testbed without noise and on a version of the virtual testbed with noise. As noise, we sample random values for each of the actuators from the dead-zone characterization in Figure 1 (c) and reduce the absolute value of the action by these sampled values. Figure 14 shows the results next to the agent pre-trained on the experiment with lower goal powers. Panel (a) shows the normalized return plotted against training steps. Overall, the agent pre-trained on the virtual testbed without noise is more stable but not significantly better or faster in training. The agent pre-trained on the virtual testbed with noise reaches higher returns and is faster than the other two. Panel (b) shows test results (time needed to fiber couple to $P_{\text{goal}} = 0.9$) for the three agents and two agents trained only in the virtual testbed, either with or without noise. The agent pre-trained on the virtual testbed with noise performed slightly better than the other two pre-trained agents, which showed no significant difference between them. The two agents only trained in the virtual testbed are significantly slower, the one trained with noise being slightly faster than the other. However, they are not as slow that it could not be useful: If the time for coupling is not relevant, it might be enough to learn on the very simple virtual testbed (even without noise).

# E Algorithm Hyperparameters

We use the default hyperparameters in StableBaselines3 (incl. contrib), Version 2.3.0 [34] or the way they appear in their tutorials. For completeness, we list them here and print the ones that are not default but used in the tutorial in bold.

TQC learning rate: 0.0003, replay buffer size: 1000000, learning starts after 100 steps, batch size: 256, soft update coefficient: 0.005, discount factor: 0.99, update model every step, do 1 gradient step after each rollout, no added action noise, update target network every 1 step, number of quantiles to drop per net: 2, number of critics networks: 2, number of quantiles for critic: 25

SAC learning rate: 0.0003, replay buffer size: 1000000, learning starts after 100 steps, batch size: 256, soft update coefficient: 0.005, discount factor: 0.99, update model every step, do 1 gradient step after each rollout, no added action noise, update target network every 1 step,

TD3 learning rate: 0.001, replay buffer size: 1000000, learning starts after 100 steps, batch size: 256, soft update coefficient: 0.005, discount factor: 0.99, update model every step, do 1 gradient step after each rollout, **action noise: NormalAction-Noise(mean=np.zeros(number actions), sigma=0.1 × np.ones(number**

**actions)**, policy and target network updated every 2 steps, standard deviation of smoothing noise on target policy: 0.2, clip absolute value of target policy smoothing noise at: 0.5

DDPG learning rate: 0.001, replay buffer size: 1000000, learning starts after 100 steps, batch size: 256, soft update coefficient: 0.005, discount factor: 0.99, update model every step, do 1 gradient step after each rollout, **action noise: NormalAction-Noise(mean=np.zeros(number actions), sigma=0.1 × np.ones(number actions)**

PPO learningrate: 0.0003, number of steps between updates: 2048, batch size: 64, number of epochs when optimizing surrogate loss: 10, discount factor: 0.99, factor for trade-off between bias vs. variance for GAE: 0.95, clip range: 0.2, normalize advantage, entropy coefficient: 0.0, value function coefficient for loss calculation: 0.5, maximum norm for gradient clipping: 0.5

A2C learning rate: 0.0007, number of steps between updates: 5, discount factor: 0.99, factor for trade-off between bias vs. variance for GAE: 1.0, entropy coefficient: 0.0, value function coefficient for loss calculation: 0.5, maximum norm for gradient clipping: 0.5, RMSProp epsilon: 1e-05, use RMSprop

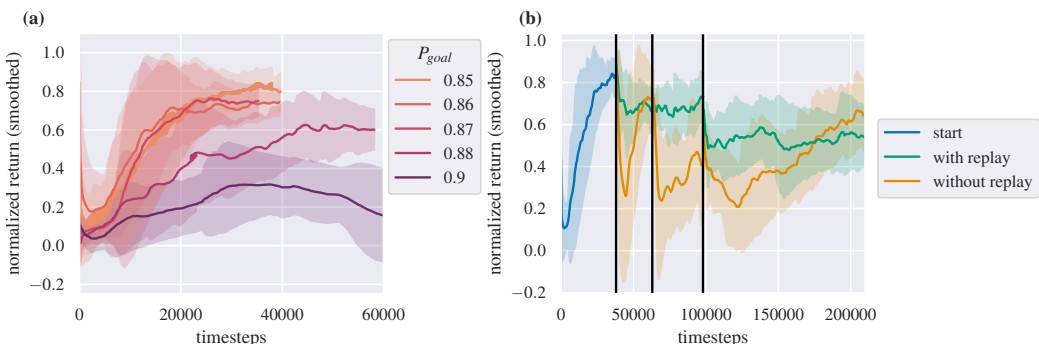

Figure 13: Both panels show the normalized return plotted against the training step. (a) shows the training from the start for different goal powers. In (b), $P_{\text{goal}}$ is raised in steps at each black vertical line from 0.85 over 0.875 and 0.89 to 0.9. For training one of the models (yellow), we delete the replay buffer after the first two black lines; for the other (green), we do not. Both panels show the mean with $2\sigma$ error bands created by smoothing.

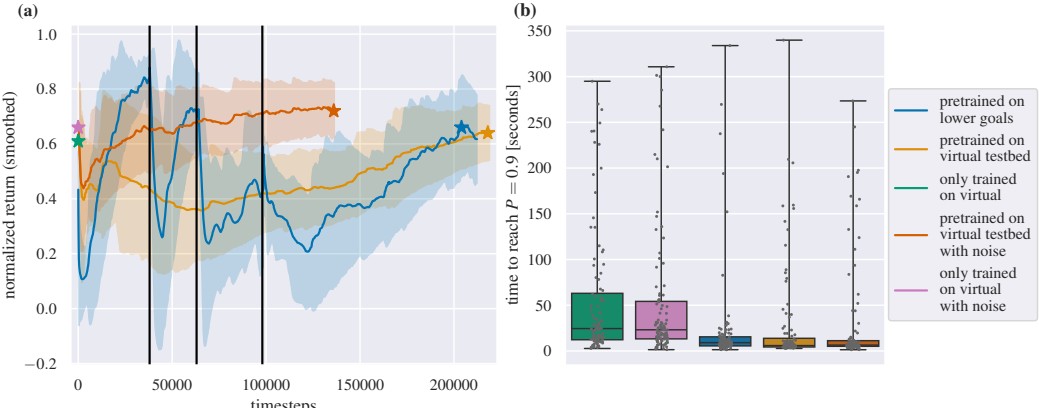

Figure 14: Pre-training on virtual testbed. (a) shows the normalized return plotted against time for three agents: one is trained directly on the experiment with successively higher goal powers (blue), the other two are already pre-trained for $5 \cdot 10^5$ training steps on the virtual testbed either without noise (orange) or with noise (red). We used $P_{\text{goal}} = 0.9$, TQC, and the parameters in Tables 1 and 2. (b) shows how long the agents marked with a star in (a) (green and pink are both only trained on the virtual testbed, without or with noise, respectively) need to couple to $P_{\text{goal}} = 0.9$ on the experiment. (a) shows the mean with $2\sigma$ error bands created by smoothing.

