# OpenReview forum: "Model-free reinforcement learning with noisy actions for automated experimental control in optics"
_ICLR.cc/2025/Conference — Submitted to ICLR 2025_

### Official Review · Reviewer_AXyW · 2024-10-20

**Soundness:** 1
**Presentation:** 2
**Contribution:** 1
**Rating:** 3
**Confidence:** 5

**Summary:**

The paper discusses using RL for automated control in optical experiments, specifically for coupling laser light into optical fibers. The authors demonstrate that an RL agent, trained directly on the experiment without pre-training, can achieve coupling efficiencies comparable to human experts. The authors use standard SAC and TQC.

**Strengths:**

1. The paper presents a real-world deployment of reinforcement learning, which I find valuable given the challenges of implementing RL on hardware.

2. The experimental setup is thoroughly detailed, offering insights for practitioners and researchers looking to deploy RL in a similar application.

**Weaknesses:**

1. My main concern is that this paper reads more like a technical report on applying standard RL algorithms to a specific problem. The application itself does not appear to be novel, and given the lack of additional contributions beyond using off-the-shelf RL techniques, I believe this paper is not relevant to ICLR. I would recommend the authors consider submitting to venues focused on optics, sensors, or robotics.

2. The authors highlight challenges in applying RL to real-world problems due to poor sample efficiency and argue that their environment is particularly difficult due to partial observability and noisy actions. These are valid challenges that could inspire algorithmic innovations to tackle them. However, the authors simply use existing model-free RL algorithms without any modifications. If sample efficiency is a concern, model-based methods would be more appropriate, or the authors can consider pretraining the policy with expert data. Similarly, if partial observability is an issue, methods designed for POMDPs or architectures with memory should be considered. Lastly, the impact of noisy actions is not clear to me, even though it is in the tilte of the paper. In my opinion, noisy actions is nothing but stochasticity in the transition dynamics, and RL algorithms are by definition applicable to stochastic transition dynamic. Given the fact that standard RL algorithms have worked in this case, the authors should either downplay these challenges or address them with appropriate methods and more thorough investigations.

3. I identified some inaccurate statements in the paper:
* L43: “Only very few [RL applications] were done in real-world environments.” This is misleading as there are hundreds of papers on RL applied to real-world robotics, beyond simulations. Additionally, RL has been applied to complex real-world tasks like fusion reactor control [1], balloon navigation [2], and data center cooling [3].
* L474: “Our experiments are the first steps towards RL in optics.” While I am not an optics expert, several papers have already applied RL in this field, some of which are cited by the authors, though others are missing [4, 5, 6]. The primary distinction in this work seems to be the direct training of the RL agent on hardware, which, albeit challenging, is not considered a novel contribution. In fact, as stated in Point 2 above, perhaps offline pretraining of the policy on expert data would be an even more suitable direction.

4. The main body of the paper includes many experimental details that are unnecessary for readers unless they intend to replicate the exact setup. This makes the paper harder to follow and obscures key information. I suggest moving these details to the appendix. Additionally, the paper contains repetitive statements, with the same points being reiterated in slightly different ways (e.g., strengths of RL algorithms, challenges in applying RL, etc.) without adding new insights. Reducing these repetitions would improve clarity.

[1] Tracey, Brendan D., et al. "Towards practical reinforcement learning for tokamak magnetic control." Fusion Engineering and Design 200 (2024).

[2] Bellemare, Marc G., et al. "Autonomous navigation of stratospheric balloons using reinforcement learning." Nature 588.7836 (2020).

[3] Lazic, Nevena, et al. "Data center cooling using model-predictive control." Advances in Neural Information Processing Systems 31 (2018).

[4] Pou, B., et al. "Adaptive optics control with multi-agent model-free reinforcement learning." Optics express 30.2 (2022)

[5] Nousiainen, Jalo, et al. "Laboratory experiments of model-based reinforcement learning for adaptive optics control." Journal of Astronomical Telescopes, Instruments, and Systems 10.1 (2024).

[6] Mareev, Evgenii, et al. "Self-Adjusting Optical Systems Based on Reinforcement Learning." Photonics. Vol. 10. No. 10. MDPI, 2023.

**Questions:**

1. I am aware of the experiments on pretraining the policy on a virtual environment and transferring it to the real setup. But that may be too challenging because of the gap between simulation and reality. However, what is the issue with pretraining the policy on expert demonstrations (from humans, or classic controllers) from the actual setup and fine tuning online on the actual setup? This can be a very reasonable approach if deploying on the actual system is expensive.
2. How does the performance of imitation learning on expert data compare to RL?
3. Are noisy actions anything beyond stochasticity in the transition dynamics? What was done specifically regarding noisy actions?
4. What do shaded regions in the figures represent?
5. How many seeds have been used in the experiments?

---

> ### Author Response · Authors · 2024-11-21
> **Answer to Reviewer AXyW, part 1**
>
> Thank you very much for taking your time to review our paper and the helpful comments that we would like to address below.
>
> About your concern whether the paper is relevant to ICLR:\
> The ICLR call for papers explicitly calls for “applications to physical sciences (physics, chemistry, biology, etc.)”. Our work falls into this category. In discussions within the community, we have learned that there is currently no consensus on how to compare the challenges across fields, e.g., between robotics and optics. The field of optics is already so diverse that experiments in the field of  astronomy (adaptive optics), for example, cannot be transferred to the tabletop laser experiments we are dealing with. In this field, the application of RL is hardly investigated especially when directly working on the experiment. We are therefore certain that our work is an important contribution to the application of RL to controlling laser optics experiments, which is part of the call mentioned above.
>
> About the remark about highlighting challenges in applying RL to real-world problems:\
> Of course, model-based algorithms are generally more sample-efficient than model-free ones. For this reason, we are currently investigating the use of model-based methods (in combination with memory-based architectures) in this environment. However, in this work we address model-free methods as their out-of-the-box usability lowers the bar of using it in other labs. Furthermore, while model-based methods would likely significantly reduce training time in the lab, their implementation requires more human effort, which may be more constrained than the time spent using the experiment.\
> While architectures with memory are a tool for dealing with partial observabitlity, so is taking a history of single observations as the observation of the environment. Again, the training time in the lab was traded off against the implementation time. We agree that noisy actions lead to nothing but a stochastic environment. However, the term has been used to clarify where the stochasticity comes from physically. In many environments, stochasticity arises from noise in the observations, which it also does in the power measurements here, but the stochasticity in the experimental fiber coupling environment is dominated by the inaccuracy of the actuators. While standard RL algorithms can work in stochastic environments, this task can be significantly harder than solving the equivalent deterministic environments. To demonstrate this, a subsection will be added to the Appendix exploring RL in environments with different levels of noise.
>
> About your remark about inaccurate statements in the paper:\
> We agree that the statement in line 43 is worded too strongly. Hence, the sentence “Only very few [RL applications] were done in real-world environments.” is changed to “Comparatively few experiments were done in real-world environments.” to make clear what is meant in the future upload. Furthermore, [1-3] will be cited additionally.\
> While we agree that our paper is not the first step of using RL in optics, the cited sentence “Our experiments are the first steps towards RL in optics.” does not appear in the paper. Line 474 reads “Our experiments are a first step towards the extensive use of RL in our quantum optics laboratory.” which is true.\
> The field of optics is very versatile, and sub-disciplines often have a low overlap.  For instance, the field of adaptive optics, for which we provided 5 citations [45-50], is highly relevant for imaging in astronomy. We included your references [4,5] in this list.  In comparison, most research and development in optics is pursued with table-top laser experiments covering fundamental physics, quantum technologies, metrology, and laser development, among others. In this area, the application of RL is relatively novel, in particular training RL on the experiment, as discussed in the related work section. Your reference [6] falls in this field and is a publication that we missed and now added. It describes the application of RL to a 1-dimensional problem to stabilize the intensity of an X-ray source. The paper is particularly relevant as it, furthermore,  tests RL on a fiber coupling task in a briefly discussed side project. Similarly, the authors identify the problem of backlash as a limitation. However, their approach is much more limited as only a discrete action space of four actions and a discrete observation space of ten observations are used. We will add this information in the related work section.

---

> > ### Author Response · Authors · 2024-11-21
> > **Answer to Reviewer AXyW, part 2**
> >
> > Concerning the remark about unnecessary experimental details:\
> > We will move some unnecessary information, such as the software and graphics card used, to the appendix. In general, our paper describes an application of RL to a specific problem and, as such, is intended to provide sufficient details to replicate the experiments, this was explicitly pointed out as positive by reviewers K3XD and nH2K. We have already reduced the content of the main text by providing an extensive appendix. The structure of the appendix allows the reader to read the paper selectively. Accordingly, we currently have difficulty identifying passages that should be shortened. Similarly, we only try to repeat content where it is common practice, i.e., abstract, introduction, and summary. However, as authors who are very familiar with the text, we may also be blind to such passages and would therefore appreciate any specific reference.
> >
> > About the question on pretraining the policy on expert demonstrations (Question 1):\
> > We fully agree that this is a very interesting point to investigate further. In this work the aim was to demonstrate how the problem of aligning a laser beam can be solved fully relying on model-free RL and therefore minimizing human investment. Requiring less human effort also makes the use of RL more accessible and increases its applicability to scientific communities that are not as familiar with RL. Implementing this idea would require a user interface to adequately control the motors manually, similar to the hand-steering mirrors, which does not yet exist. Therefore, this is not feasible for this work and was not the goal. However, we are eager to investigate this in future work!
> >
> > About the question on imitation learning (Question 2):\
> > Imitation learning on expert data requires expert data in the same way as pre-training on expert data. As discussed in Question 1, this expert data cannot be easily obtained in our current setup. However, this would be an interesting question for future work.
> >
> > About the question whether the noisy actions are anything beyond stochasticity in the transition dynamics (Question 3):\
> > As explained in the reply to Weakness 2, noisy actions are only a specific case of stochasticity in the environment, which will be made clearer in the paper. TQC is an algorithm specifically learning reward distributions instead of expected rewards (Q-values), which is especially interesting in stochastic environments.
> >
> > About the question on what do shaded regions in the figures represent (Question 4):\
> > The shaded regions are 2 $\sigma$ error bands as mentioned in lines 366-669. This was only mentioned in the text , but will be added to the figure caption.
> >
> > About the question about seeds (Question 5):\
> > In each experiment in the virtual testbed, five seeds have been used. However, as they have not been set explicitly, they are not comparable across experiments. This issue will be addressed for the camera-ready version (fully addressing it during the rebuttal is unfortunately not possible). In the lab, due to higher training times, only for the comparison of SAC and TQC two seeds were used, the other experiments only include one run.

---

> > > ### Comment · Reviewer_AXyW · 2024-11-21
> > >
> > > Thank you for your rebuttal.
> > > * **Relevance to ICLR:** While there is indeed a call for applications, I believe this is meaningful only if the work demonstrates sufficient novelty and contributions in one or more of the following areas:
> > >     * Theoretical contributions
> > >     * Algorithmic contributions
> > >     * Novel application (e.g., addressing a previously unsolved real-world problem, or applying machine learning to a new domain for the first time)
> > >
> > >     Based on my understanding, this paper does not fall into any of these categories, as I also mentioned in my review. Please do not get me wrong—I do value hardware implementation, as highlighted in the strengths of my review. However, I think this work does not offer enough contributions beyond applying off-the-shelf RL implementations, which limits its relevance to ICLR, as also noted by **nH2K**.
> > >
> > > * **Impact of noisy actions:** I remain unconvinced by the response provided. I still believe that the focus on noisy actions should be reduced in the paper, as standard RL methods have demonstrated success in handling such scenarios effectively.

---

### Official Review · Reviewer_nH2K · 2024-11-02

**Soundness:** 2
**Presentation:** 3
**Contribution:** 1
**Rating:** 5
**Confidence:** 4

**Summary:**

An applied contribution proposing the use of off-policy continuous action space RL algorithms (SAC, TQC, TD3, etc.) for controlling four stepper motors tilting two mirrors in optics experiments. The goal is to automate the alignment of a laser beam for coupling to an optical fiber.

In this setup the main algorithms tested appear to perform more or less the same, with the only difference being made by the value of the target power.

**Strengths:**

- good exposition of the problem being solved and the associated challenges
- extensive experiments demonstrating the feasibility of the application
- detailed description of the experimental setup. Without a background in optics experimentation I got the feeling that I could reproduce the experiments to some degree if required to. The paper contains thorough descriptions of design choices regarding the actions pace, processing of observations, formulation of the reward function, reset procedure, etc.

**Weaknesses:**

The only reserve I have is **whether ICLR is a good venue for this submission**. This contribution demonstrates that off-the-shelf RL algorithms (literally StableBaseline implementations) can be used for automating parts of the calibration and setup of optics experiments in the presence of noisy actuators.

This in itself is not surprising nor novel. DRL algorithms have been shown to perform control in real-life experiments with noisy actuators, partial observability and, in addition to this work, noisy observations, multiple times and even for higher-dimensional problems. In its current form the paper presents no findings that can be of general interest for the broader RL community.

It would probably be an excellent contribution to a more applied venue.

**Minor:**
- figures presenting multiple algorithms and power goals in the same pane can be confusing / difficult to follow (eg.: fig 2a-b).

**Questions:**

- Why evaluate only TQC in fig 2e and not SAC also?
- What is the interpretation of fig 2d? Why is the number of steps of the last episode important?

---

> ### Author Response · Authors · 2024-11-21
> **Answer to Reviewer nH2K**
>
> Thank you very much for taking your time to review our paper and the helpful comments that we would like to address below.
>
> Regarding the reserve whether ICLR is a good venue for this submission:\
> The ICLR call for papers explicitly calls for “applications to physical sciences (physics, chemistry, biology, etc.)”. Our work falls into this category. In discussions within the community, we have learned that there is currently no consensus on how to compare the challenges across fields, e.g., between robotics and optics. The field of optics is already so diverse that experiments in the field of astronomy (adaptive optics), for example, cannot be transferred to the tabletop laser experiments we are dealing with. In this field, the application of RL is hardly investigated especially when directly working on the experiment. We are therefore certain that our work is an important contribution to the application of RL to controlling laser optics experiments, which is part of the call mentioned above.
>
> Regarding the minor comment about presentation issues in Fig 2a-b:\
> We will address this in a future upload of the paper by normalizing the return.
>
> Regarding the question why Fig 2e does not show an evaluation for SAC:\
> In Figure 2 (e) only the agents trained on a goal power of $P_\text{goal}=0.9$ are shown. Due to limited time SAC and TQC were only compared against each other for $P_\text{goal}=0.85$. For higher goals, only TQC was used. Hence, SAC is not shown in Figure 2 (e). We can work on incorporating SAC also for $P_\text{goal}=0.9$ and then show the results in Figure 2 in the camera-ready version.
>
> Regarding the question about the interpretation of Fig. 2d:\
> The goal of this plot is to answer the question: in an episode where the agent is successful, how many steps does it take to reach the goal? In most cases (83% to 97%, depending on the goal power) the agent reaches the goal in the first episode. However, in some cases, it requires more than one reset to reach the goal. We wanted to show here how many steps it takes in the successful episode (often the first, but always the last).

---

### Official Review · Reviewer_oBMD · 2024-11-08

**Soundness:** 4
**Presentation:** 3
**Contribution:** 4
**Rating:** 8
**Confidence:** 4

**Summary:**

The coupling of laser light into optical fibers is a difficult task in optics labs because of its high precision and sensitivity requirements. This work investigates the use of model-free reinforcement learning (RL) to automate and optimize this process. The work shows that RL can effectively handle motor imprecision and experimental noise without the need for a comprehensive simulation model. By employing the Truncated Quantile Critics (TQC) and Soft Actor-Critic (SAC) algorithms, the authors get coupling efficiencies above 90%, which is on par with human experts. By demonstrating RL's promise in automated optical alignment tasks, this work opens the door for its application in intricate optical experiments where comprehensive noise modeling or conventional control techniques are impracticable.

**Strengths:**

Originality: In optical systems, where simulation is difficult due to noise and system complexity, this research uses model-free reinforcement learning in an area where traditional models fall short.

The RL agent's ability to attain high coupling efficiencies directly on physical hardware, matching or surpassing human performance, is demonstrated by rigorous experimental results.

Significance: The effective use of RL in this context may encourage the development of comparable methods in other physical and optical disciplines where intricate, noisy situations defy conventional modeling.

Clarity: Detailed experimental methods improve reproducibility, and the report clearly conveys its approach and the justification for algorithm selections.

**Weaknesses:**

Limited Generalization: Although the RL agent performs admirably, it is designed especially for this particular experimental configuration. The significance of the paper would be enhanced by greater generalizability or adaptability to different optical settings.

Training Time: The practical implementation of this strategy in time-sensitive settings may be limited by the time needed to reach high efficiency, which can take up to four days.

Dependency on Particular Hardware: The configuration is dependent on motorized mirrors and particular power measurements, which may restrict its direct application to other labs without comparable apparatus. Investigating different sensing techniques may increase flexibility.

**Questions:**

Could transfer learning from pre-trained models or virtual simulations speed up the training process?

How resilient is the RL agent to modifications in the optical configuration, such as slight misalignments or shifts in the location of the optical fiber?

Could the method be applied to tasks with more complicated noise distributions or higher dimensional control requirements?

Given the lengthy training procedures, is the agent's performance sustainable in terms of wear on mechanical components?

---

> ### Author Response · Authors · 2024-11-21
> **Answer to Reviewer oBMD, part 1**
>
> Thank you very much for taking your time to review our paper and the helpful comments that we would like to address below.
>
> Regarding the concern about limited generalization: \
> Could you specify what kind of generalization you mean? Do you mean applying the trained agent to a different setup or using our method for a different experiment?\
> Regarding the first option: in principle a trained agent should be able to handle other beam alignment tasks. The maximum allowed action ($a_{max}$) would definitely need to be adjusted as this depends on the lengths between the different optical components. Furthermore, the agent would probably need some extra training to learn the characteristics of different actuators as this is different for each individual device. We will not be able to show experimental results from implementing the agent on a different setup during the rebuttal period but we are working on it.\
> Regarding the second option: there is an explanation in lines 462-466 of how this generalizes to any beam alignment task. Please let us know if there is anything unclear there and what would be needed for better understanding.\
> Additionally, and this is related to question no. 2 (resilience of the agent), the agent is  resilient to changes in the optical configuration such as slight misalignment: Lines 431-439 and Table 3 show that the agent is able to realign the experiment achieving high output powers after a misalignment occurred at a location inaccessible  to the agent (whether this misalignment happened at the location of the fiber or at another element not accessible to the agent is equivalent in terms of difficulty). We consider this resilience to small misalignments or shifts to be a major achievement and apologize if this was not directly clear from the paper. The equivalence of the investigated misalignment to slight shifts in the location of the optical fiber will be explicitly mentioned in a future upload.
>
> Regarding the concern about training time:\
> The goal was to find a strategy that would minimize the time and effort required by the human experimenter. Although shortening the training time in the lab was a goal, the main interest was to do this without sacrificing more human time, especially since the training can be done over night or the weekend when the lab is not used. However, a different choice can be made: it is possible to shorten the training time in the lab by putting more effort into pre-training.
>
> Regarding the concern about dependency on particular hardware:\
> To automate mirror movements, motorized mirror mounts are a must. However, the motors do not have to be exactly the ones used in this work. Moreover, since it has been demonstrated that this method works with actuators with fairly unpredictable noise, we are confident that many other actuators can full fill the job, even self built ones. About sensing: This work used power meters. But regular photodetectors could be used instead. The photodetector would give, similarly, a scalar proportional to the power which is all that is needed. The presented sensing technique is simple. Any other sensing technique, e.g. to obtain beam orientation information, would be much more sophisticated and disproportionate to its benefit.

---

> > ### Author Response · Authors · 2024-11-21
> > **Answer to Reviewer oBMD, part 2**
> >
> > Regarding the question about transfer learning:\
> > As explained in Appendix D.3 (lines 1368-1383), pre-training on the virtual testbed including noise can cut training time in the lab in half. However, this requires a noise model of the actuators that must first be built in the lab, which also takes time. Furthermore, as explained in lines 401-402, using models pre-trained on lower goals helps in the sense that higher returns are achieved with the same amount of training time. In turn, this means that without pre-training, agents need at least  more training time to achieve the same return values.
> >
> > Regarding the question about the resilience of the RL agent:\
> > Please see the answer regarding the concern about limited generalization, especially the last paragraph, in part 1 of our answer.
> >
> > Regarding the question about more complicated noise distributions:\
> > Did you mean different noise levels of the same kind of distribution or different kinds of noise distribution? In an upcoming upload of the paper, a section on different levels of noise will be added. From preliminary results, for $P_\text{goal}=0.85$, it is clear that the agent still learns with higher noise, but it takes longer and performs a little worse. If you mean different kinds of noise distributions, what would you have in mind (except for a distribution similar to the one in the lab or a Gaussian distribution)?
> >
> > Regarding the question about higher dimensional control requirements:\
> > In general, the higher dimensional the action-space, the longer the training time in a similar environment. For example, an agent that can only move one or two actuators trained in an environment where only these actuators are misaligned, learns much faster (only taking 1.5% or 21.5% of the time) than one trained to move all four actuators. Hence, for more actuators, either the training time will be longer, pre-training gets more important, and/or more optimization has to happen in the virtual testbed. However, using four actuators (two mirrors) and one sensor is enough for the spatial alignment of light. For more complicated spatial alignment tasks, i.e., if the light has to be aligned at several points, it would make more sense to train several agents on mirror-mirror-sensor blocks than to add more actuators to the one agent.
> >
> > Regarding the question of wear on mechanical components:\
> > This is an important question. In the selection  of the motors, care has been taken to ensure  that they have a long life time. So far, we could not observe any indication of wear. It is assumed that if training increases in more complicated experiments this effect might become more relevant and  using pre-training or pre-learning from a human expert becomes more important to reduce training time. However, we assume that the biggest effect of wear in the components is a change in the imprecision of the actuators. Therefore, wear of these components will change the noise in the actions. Through additional training on the experiment, the agent can therefore learn to deal with a changed noise situation. We will investigate if there is a change in motor behavior in future work.

---

> ### Comment · Reviewer_oBMD · 2024-11-21
> **Response to the comments**
>
> Thank you for your thoughtful responses, which address many concerns effectively. However, several areas could still be improved to strengthen the paper. Below are my concise thoughts as a critical reviewer:
>
> Generalization
>
> Your explanation of generalization is helpful, but stronger empirical or theoretical support is needed:
>
> - Across Setups: The claim of retraining and adjusting actions for new actuators would benefit from experimental validation or detailed guidelines on required modifications.
>
> - To Other Experiments: Conceptual arguments about equivalence in beam alignment tasks lack supporting data. Adding results from even one additional experiment, simulated or real, would substantiate this claim.
>
> - Training Time: The argument that training time offsets human effort is reasonable but not universally persuasive. Reducing total experimental time remains critical, especially when setup access is limited. The benefits of pre-training, as noted in Appendix D.3, should be emphasized more in the main text, along with a clear comparison of noise model construction time versus direct training.
>
> - Hardware Dependency: Your clarification on flexibility in hardware and sensing methods is valuable but should be more explicitly highlighted. Demonstrating alternative sensing setups (e.g., photodetectors) would improve accessibility and adaptability.
>
> - Transfer Learning: While the benefits of pre-training are clear, the time required to build a noise model remains ambiguous. Quantifying this effort and comparing it to saved training time would make the efficiency gains more compelling.
>
> - Resilience and Noise Distribution: The agent’s resilience to misalignments is a significant achievement but underemphasized. Explicitly linking this to practical scenarios would strengthen the impact. For noise distributions, the discussion remains abstract—concrete examples (e.g., thermal drift or non-Gaussian noise) would be helpful.
>
> - Higher-Dimensional Control: Your explanation of the tradeoff between action-space dimensionality and training time is clear, but alternative strategies like policy decomposition or hierarchical training could be explored. Including experiments with more than four actuators would broaden applicability.
>
> - Mechanical Wear: While the explanation on wear is plausible, the absence of long-term testing weakens scalability claims. Including preliminary data or plans for durability testing would strengthen this section.
>
> Suggestions
> - Generalization: Provide case studies or experimental validation for adapting agents to new setups.
> - Training Time: Highlight pre-training benefits and quantify noise model development versus direct training.
> - Hardware Versatility: Showcase alternative setups with additional data or demonstrations.
> - Resilience: Expand on real-world examples of misalignment resilience.
> - Higher Dimensions: Explore and validate scaling solutions like policy decomposition.
> - Wear Testing: Include plans or data on long-term mechanical wear.
>
>
> Conclusion
>
> While many concerns are addressed, some responses rely too heavily on future work. Strengthening empirical support for generalization, hardware adaptability, and training time efficiency would elevate the paper from a promising demonstration to a robust, broadly applicable contribution.

---

> > ### Author Response · Authors · 2024-12-03
> >
> > Thank you for your time and feedback. Unfortunately, most of these suggestions are too time-consuming to address in the discussion period. We will nevertheless use these points for future work.

---

### Official Review · Reviewer_K3XD · 2024-11-09

**Soundness:** 4
**Presentation:** 3
**Contribution:** 2
**Rating:** 5
**Confidence:** 4

**Summary:**

This paper presents an application of deep reinforcement learning algorithms to a physical task of optical fiber coupling. As designed by the authors, the RL agent controls two motorized mirrors, each with two degrees of freedom. The task is to move the mirrors to align the laser beam for the highest output power.

The control problem emphasizes practicality. Concretely, the authors employ a POMDP setting where the observations only consist of fields related to the measurable power output and historical actions. This design choice ensures the policy wouldn't rely on mirror position readings (subject to drift) and other hard-to-measure metrics. As a result, a policy is learnable directly on real hardware. The paper also details the experiment design, including the reward function, resetting scheme, and curriculum learning.

Experiment results show that the proposed RL framework derives a control policy that performs the optical fiber coupling task faster than a human expert.

**Strengths:**

- The authors have rigorously designed the experiments, performing method selection in simulation and experimentation on hardware.
- The presentation is very clear. The paper covers the problem of interest, potential challenges, design choices, experiment designs, and results in great detail.
- The main experiment results are convincing: the RL agent can overcome action stochasticity and perform from the limited observations to solve the optical fiber coupling problem above the human level.

**Weaknesses:**

- It is nice to see RL applied to a physical control system and have utilities for research in another domain. However, the control problem solved in the work is relatively simple, especially when the robotics community has trained RL policies on higher degrees of freedom systems, even from image observations.
- I sense there's not much representation-learning in this project. I think the insights will be more relevant if the authors can scale the experiment up to higher-dimensional observations. For example, is there a way to have a visual observation of laser paths as input to the policy?

**Questions:**

- Regarding Figure 2, my understanding is that the reward function differs for different goal power levels. In this case, plotting episode returns in the same figure is not the most appropriate. Is there a way to define a normalized or common metric?
- Prior work in RL has shown that a few human demonstrations can help speed up RL, even overcoming the sparse reward problem. I wonder if using a few human demos can accelerate learning and maybe even remove the need for the dense reward function.

---

> ### Author Response · Authors · 2024-11-21
> **Answer to Reviewer K3XD**
>
> Thank you very much for taking your time to review our paper and the helpful comments that we would like to address below.
>
> The two listed weaknesses mainly target the dimensionality of the actions and observations.  Specifically in the field of  table-top laser experiments, to our knowledge, there are no applications were RL has been trained on hardware with higher action spaces/more degrees of freedom. Hence, the novelty lies primarily in the application. Although a higher dimensional observation would be more interesting from a representation learning perspective, it is not reasonable for the given example. Indirect imaging, as used in robotics, is not possible due to the low scattering of laser light in air. Instead, additional beam splitters and (expensive) sensors are needed to probe the beam directly at different locations. It remains to be seen whether such sensors would improve the performance of our agent. However, in our opinion, they are not proportionate to the effort involved. For future experiments, such as optimizing the interference between two beams, beam imaging could provide additional valuable information.
>
> Thank you for the question regarding Figure 2. We decided to normalize the return by dividing it by the maximum possible return, which leads to better comparability for different goals. This will be changed in the upcoming version of our manuscript.
>
> Regarding the question about human demonstrations:\
> We fully agree that this is a very interesting point to investigate further. In this work the aim was to demonstrate how the problem of aligning a laser beam can be solved fully relying on model-free RL and therefore minimizing human investment. Requiring less human effort also makes the use of RL more accessible and increases its applicability to scientific communities that are not as familiar with RL. Implementing this idea would require a user interface to adequately control the motors manually, similar to the hand-steering mirrors, which does not yet exist. Therefore, this is not feasible for this work and was not the goal. However, we are eager to investigate this in future work!

---

> > ### Comment · Reviewer_K3XD · 2024-11-22
> >
> > Thanks for your response! I now understand that higher-dimensional sensory observation might not be wise in this application.
> >
> > As I wrote in the original review, I appreciate a working cross-domain application of RL. However, I share the same concerns with other reviewers about its relevance to this venue. Hence, I'll keep my current rating.

---

### Author Response · Authors · 2024-11-28
**Changes to the paper**

Thank you for your comments. We made the following changes throughout the paper based on your reviews. We also performed additional experiments in the virtual testbed studying the impact of different noise levels, which can be found in the newly added Appendix C.8.

Logs of the changes:

All Figures: The information that the shaded areas are 2 $\sigma$ error bands/bars was included in the caption of the figures instead of the text (formerly L366-369). (Reviewer AXyW, Q4)

All Figures: normalized return throughout the paper by dividing the return by its maximum possible value for each goal power to make it comparable across goal powers (Reviewer K3XD, Q1; Reviewer nH2K W4)

L43-44: “Only very few [RL applications] were done in real-world environments.” → “Comparatively few experiments were done in real-world environments.” (Reviewer AXyW, W3)

L58: added that noisy actions are a special type of stochasticity (Reviewer AXyW, Q3, W2)

L85:  added more citations for adaptive optics applications (Reviewer AXyW, W3)

L99-105: added citation and explanation of X-ray experiment and comparison to our work (Reviewer AXyW, W3)

L146-147: Changed the sentence formerly in L144 to make clear that two mirrors each tiltable in two axes are sufficient for any beam alignment, added citation (Reviewer oBMD, W1)

L220-222: added that noisy actions are a special type of stochasticity, and refer to Appendix C.8 for more information on the influence on the learning behaviour  (Reviewer AXyW, Q3, W2)

L313-316: added explanation of normalized return (Reviewer K3XD, Q1; Reviewer nH2K W4)

L373: added to explanation of Figure 2 (d) that the episode after the last reset is the succesful episode (Reviewer nH2K, Q2)

L458-460: added the following into the description of using the agent for correction of slight misalignment: A misalignment at the hand steering mirrors is equivalent to a slight shift in the location of the fiber collimator (or any other element not accessible to the agent) (Reviewer oBMD, Q2)

L495-496: added that future work should include investigations on whether pretraining on expert demonstrations could speed up the learning process (Reviewer oBMD, Q1; Reviewer K3XD, Q2)

L511-515: added additional last paragraph to Summary and Outlook to clarify the aim of the paper

L930-932: details on graphics card and used packaged moved from L364 to Appendix A (Reviewer AXyW, W4)

L1382-1397, L1440-1457: added Appendix C.8  with experiments for different levels of noise to explain its influence on the agent (Reviewer AXyW, Q3, W2, Reviewer oBMD, Q3)

The supplementary material was updated accordingly (noise experiments, normalized return).

---

### Author Response · Authors · 2024-12-03
**General Clarifications**

We want to thank all reviewers again for their respectful feedback. In this general response, we would like to clarify two aspects:

We were asked several times to provide additional experimental validation. Conducting experiments in an optics lab can be very time-consuming and financially costly. It is, therefore, not feasible to quickly conduct more experiments (please note that also the email about the extension of the rebuttal period stated that no additional significant experiments should be requested).

The use of RL is still very novel in the laser optics community, with very few examples of successful applications to actual experiments in the lab. However, we are sure that RL provides great potential for this community but is often not easily accessible. The main aim of our work is to show that common optics tasks can be solved and streamlined using existing RL techniques, avoiding the need to dive deep into algorithm development or build sophisticated simulations. Several reviewer’s comments point out ways to possibly improve/optimize our approach and we fully agree (as written in our Summary and Outlook section). However, as the application of RL in this domain is novel, we aimed to demonstrate that it is possible to use RL for a common optics task rather than showing the best and most general way.

Thanks again for your time.

---

### Meta-Review · Area_Chair_GVJP · 2024-12-14

**Metareview:**

This paper proposes an application of model-free RL, using soft actor-critic and truncated quantile critics, to automate laser alignment tasks in optical systems. This is primarily an application paper on a novel domain, at least for the ICLR community. They train their agent on real hardware from scratch and compare performance to human experts in coupling laser light into optical fibers.

It is commendable and impressive to see a custom hardware implementation built from the ground up and applied to reinforcement learning. This problem is particularly challenging because of partial observability and imprecision in the hardware system. These challenges are their primary motivation for using an RL setup. The application area is certainly novel for the ICLR community and the experimental rigor is detailed. The trained agent also handles noisy actions and motor imprecisions and should have high relevance to optics researchers.

However, one of the key concerns was limited novelty for the ML community. The paper does not propose a new algorithm or learning paradigm, which is fine for an empirical paper but the setup is not shown to generalize which is typical for application papers. This is harder to do due to hardware limitations and therefore inherently more time consuming. Perhaps the bigger issue is lack of direct relevance to ICLR. I personally think this is a very interesting application and we should be building world models with model-free and model-based methods for novel domains. However, the bar for such an application paper should be to demonstrate strong baselines with clear metrics, or show generalization to perturbations / different setups, or solve a previously unsolved applied problem. So although this experimental work can and will have implications for optics researcher, it lacks broader contribution to the ML community. So I encourage the authors to continue this work and address the issues raised during the review process.

**Additional Comments On Reviewer Discussion:**

The key issues raised during the review process were: 1) Relevance to the ICLR community, 2) novelty and contributions and 3) generalization to setup perturbations.

For 1), authors argued that the paper aligns with ICLRs call for applications to physical sciences but the reviewers were unconvinced and there was lack of consensus regarding novelty or details that could be applied for other domains. I do agree with these concerns but I also want to encourage such applied papers. In my opinion, the authors should setup stronger baselines, do generalization experiments and explore the frontier of off-the-shelf RL methods to map out the frontier for this domain. This would lay out stronger foundations for other optics researchers to build upon this work within the ICLR context. I do acknowledge that the training times are long due to hardware requirements but a framework to share data with other researchers and also setup simulation platforms seem like a good direction to lay out a foundation for this work.

Another point of concern is that the reviewer with highest rating (oBMD) and the authors did not get resolution during the review process. This was called out by the authors themselves. So there was an increasing amount of lack of consensus as the rebuttal processes unfolded.

---

### Decision · Program_Chairs · 2025-01-22

Reject